# Endovascular Drug Delivery

**DOI:** 10.3390/life14040451

**Published:** 2024-03-28

**Authors:** Claudiu N. Lungu, Andreea Creteanu, Mihaela C. Mehedinti

**Affiliations:** 1Department of Functional and Morphological Science, Faculty of Medicine and Pharmacy, Dunarea de Jos University, 800010 Galati, Romania; mihaela_hincu10@yahoo.com; 2Department of Pharmaceutical Technology, University of Medicine and Pharmacy Grigore T Popa, 700115 Iași, Romania

**Keywords:** atherosclerosis, drug delivery, endovascular therapy, lesion morphology, lesion-specific treatment

## Abstract

Drug-eluting stents (DES) and balloons revolutionize atherosclerosis treatment by targeting hyperplastic tissue responses through effective local drug delivery strategies. This review examines approved and emerging endovascular devices, discussing drug release mechanisms and their impacts on arterial drug distribution. It emphasizes the crucial role of drug delivery in modern cardiovascular care and highlights how device technologies influence vascular behavior based on lesion morphology. The future holds promise for lesion-specific treatments, particularly in the superficial femoral artery, with recent CE-marked devices showing encouraging results. Exciting strategies and new patents focus on local drug delivery to prevent restenosis, shaping the future of interventional outcomes. In summary, as we navigate the ever-evolving landscape of cardiovascular intervention, it becomes increasingly evident that the future lies in tailoring treatments to the specific characteristics of each lesion. By leveraging cutting-edge technologies and harnessing the potential of localized drug delivery, we stand poised to usher in a new era of precision medicine in vascular intervention.

## 1. Introduction

Percutaneous and endovascular interventions have transformed atherosclerosis treatment, especially when combining mechanical devices, such as stents, with local drug release. Drug-eluting stents (DES) and drug-coated balloons (DCBs) address hyperplastic growth and in-stent restenosis, allowing focused treatment for de novo disease and reducing post-interventional restenosis. The key factor influencing therapeutic efficacy is the interaction between device delivery mode, drug uptake, and lesion morphology [1].

Despite progress in vascular biology, bioengineering, and pharmacology, restenosis remains challenging in vascular reconstruction. Intimal hyperplasia’s complex pathophysiology has led to the identification of drugs and tools for prevention. Optimizing local delivery faces challenges due to the innovations and complexity of modern stent designs [2]. Tissue distribution after stent delivery mirrors stent-coating geometry, affecting drug diffusion and potentially causing peak concentrations and toxicity near stent struts. Addressing this requires altering drug elution rates or engineering strut shapes, demanding a sophisticated approach. Stent-based delivery can achieve uniformity by intentionally varying drug loading or deploying drug-loaded coatings or particles in inter-strut zones [3].

Uniform endovascular drug delivery alone does not guarantee effective transmural distribution unless the delivery duration is sufficiently extended. The necessary duration for optimal arterial distribution rises with enhanced endothelial integrity and resistance to drug absorption. This requirement tends to be higher for more significant drugs with lower tissue diffusivities and greater steric retardation. Additionally, drug charge and lipophilicity play crucial roles in this process. Further, even when a drug saturates the arterial wall, its clearance can hinder therapeutic effectiveness [4]. For instance, the rapid tissue clearance of heparin, delivered via balloon-based or catheter-based methods, contributed to high restenosis rates due to its aqueous solubility. Hydrophilic molecules, such as heparin, tend to distribute more into blood than tissue and, within the tissue, reside in extracellular spaces. Soluble drugs’ uptake and clearance rates often scale with their diffusion coefficient, which can be prolonged using high-molecular-weight or charged analogs [5,6,7].

Drug-coated balloons (DCBs) offer a potential solution, primarily for ISR but increasingly as an alternative to DES for de novo lesions. Concerns include the need for a short drug delivery window, higher drug loading for DCBs, and safety issues with paclitaxel-coated DCBs [8].

Furthermore, in a minimally invasive procedure, embolization intentionally blocks diseased blood vessels for treatment. SEM, made from processed silk fibroin proteins and charged nano-clay particles, is visible and injectable through small clinical catheters. In vitro, SEMs loaded with labeled albumin and Nivolumab show sustained release over 28 days. SEMs successfully embolize arteries without recanalization in a porcine renal model, delivering albumin and Nivolumab into the renal cortex. SEM, disrupting the internal elastic membrane, proves a promising multifunctional embolic agent for treating vascular diseases, including tumors. Despite its success in some cancers, immunotherapy faces challenges in solid tumors due to immunosuppressive microenvironments and inherent barriers [9]. Current delivery methods, involving frequent intravenous administration, are costly and induce systemic toxicity. An alternative approach, combining direct immunotherapy delivery to tumors with embolization-induced local cell death, could enhance clinical outcomes, reduce toxicity, and lower costs [10].

Transcatheter arterial embolization (TAE) employs a catheter to navigate through the vasculature, delivering embolic agents directly to target tissue for intentional vessel blockage. TAE, commonly used for liver lesions, faces challenges in improving the five-year survival rate. Despite attempts with microbeads and drug delivery, challenges persist. Injectable hydrogels, such as silk-based shear-thinning hydrogel (SEM), offer advantages, combining solid and liquid characteristics for versatile embolization and therapeutic delivery. Silk fibroin, derived from silkworms, provides a biocompatible and controllably degradable matrix, enabling tunable drug release. SEM, a composite of regenerated silk fibroin gels and nano-clay, overcomes intravenous drug delivery limitations, serving as a catheter-injectable embolic material and sustained drug reservoir. Radiopaque SEM, formulated with iohexol, facilitates visibility on imaging modalities. In vivo experiments in rats and pigs demonstrate SEM’s embolization capabilities and drug delivery potential, including studies with Alexa-594-labeled bovine serum albumin, indocyanine green, and Nivolumab [11,12].

This review explores how atherosclerotic morphology impacts drug retention, clarifies the role of complex lesion characteristics in device efficacy, and envisions a shift toward lesion-specific treatment for enhanced future care [13].

A systematic electronic literature search using PubMed was performed for all accessible published articles. Additional searches were conducted for abstracts presented at relevant societal meetings, clinical trials, and funded NIH studies.

Overall, this review is focused on endovascular drug delivery methods and devices used in atherosclerotic disease therapy. Firstly, intraluminal drug delivery devices are discussed. A short overview of atherosclerotic diseases and the hyperplastic vascular response is discussed. The impact of vascular and atherosclerotic structures on drug retention is also presented. Bioresorbable stems are given special attention. Porous and microporous balloons are presented, and some commercially available devices are named. The mechanisms controlling local drug release are synthesized. The device-based endovascular drug delivery strategies are detailed. Additionally, devices for vascular drug delivery and the release mechanism are discussed. First-generation and second-generation DES (drug-eluting stents) are overviewed. Bioresorbable scaffolds, drug-coated balloons, transition toward lesion-specific drug delivery, deployable coatings, and polymer-free coated stents are presented. The discussion section includes an overview of the clinical implications of lesion-specific intervention. Furthermore, lesion-specific intervention and future drug delivery, looking toward the horizon, are presented. Lastly, extraluminal drug delivery devices, perivascular biomaterials, systemic drug administration, and targeted intravenous drug delivery are described.

### 1.1. Intraluminal Drug Delivery Devices

Drug-eluting stents (DES) are small, mesh-like tubes inserted into narrowed or blocked blood vessels to restore blood flow. They work by combining the mechanical support of a traditional stent with a drug delivery system. The stent scaffold helps keep the artery open while the drug, typically an antiproliferative or anti-inflammatory agent, is released gradually from the stent coating into the surrounding tissue. This drug helps prevent the re-narrowing of the artery, known as restenosis, by inhibiting excessive tissue growth or inflammation that can occur in response to the stent placement. By combining mechanical support with targeted drug delivery, DES effectively reduce the risk of restenosis and improve long-term outcomes for patients undergoing coronary or peripheral vascular interventions. DES have been widely used to treat coronary artery disease since the FDA approved the first generation in 2003. In the short term, DES effectively reduce intimal hyperplasia or restenosis compared to bare-metal stents (BMS). Sirolimus and paclitaxel DES significantly decrease target lesion revascularization rates, with sirolimus showing a more robust reduction—Abbott, Boston Scientific, and Medtronic market FDA-approved DES [14,15].

However, DES have limitations, including the potential for late in-stent thrombosis requiring dual-antiplatelet therapy. Drawbacks include the lack of reendothelialization, uneven drug delivery to the vessel wall, and the concentration of the drug at stent struts. Due to increased trauma to the vessel wall, stents may induce intimal hyperplasia, surpassing the restenosis degree seen with balloon angioplasty. Importantly, DES are costly, raising concerns in a healthcare era with constrained resources [16,17].

Despite the success of drug-eluting stents (DES) in coronary applications, their effectiveness in peripheral circulation has only recently become apparent. Trials, such as SIROCCO and STRIDES for superficial femoral artery (SFA) lesions, did not show significant differences between DES and bare-metal stents. However, Cook Medical reported favorable outcomes with its Zilver PTX polymer-free paclitaxel-eluting nitinol DES in treating femoropopliteal lesions [18]. The DESTINY trial demonstrated improved patency rates with Abbott’s Xience^®^ Prime everolimus-DES in infra-popliteal lesions. Ongoing trials, including PADI and ACHILLES, aim to assess the efficacy of DES for infrainguinal disease. While the dual-drug-eluting stents (DDES) and covered stent innovations show promise, clinical studies have not demonstrated significant benefits beyond bare-metal stents. Further evaluation is needed to determine the role of DES across the full spectrum of infrainguinal pathology [19,20,21]. Some intraluminal drug delivery devices are listed in Table 1, where the major intraluminal devices are briefly described together with their uses.

The hyperplastic vascular response is critical and affects the efficacy of all intraluminal drug delivery devices, and this issue will be discussed in the next section.

### 1.2. Atherosclerotic Disease and Hyperplastic Vascular Response

Atherosclerosis presents passive structural barriers affecting luminal flow and active metabolic elements with significant drug metabolism potential. Understanding various plaque phenotypes and lesion progression stages is crucial for drug delivery due to the disease’s inflammatory nature. The nature of lesions is diverse, and their advancement to an advanced, susceptible stage is influenced by various factors [32,33,34].

Local administration relies on the formation of new lesions and the vascular reaction associated with the introduction of the device. For instance, in the context of an angioplasty balloon, the expanded device scrapes the endothelium, exposes internal plaque dissection layers, and initiates immediate thrombosis and a hyperplastic response [35]. The process becomes even more intricate when permanent devices are inserted, as stent struts lead to local thrombosis, inflammation, proliferation of smooth muscle cells, and vascular restructuring.

It is also important to stress the role of the rheologic and hemodynamic environment, where modified wall shear stresses at an interventional site accelerate pathologic restenosis [36,37].

Some factors implicated in hyperplastic vascular response, mainly related to restenosis after angioplasty or stenting, are listed in Table 2, where the critical factors implied in the hyperplastic vascular response are presented together with their implications in restenosis.

Next, we present a discussion of the hyperplastic vascular response that impacts the role of atherosclerosis in endovascular drug delivery.

### 1.3. The Impact of Vascular and Atherosclerotic Structures on Drug Retention

The concept that the vascular structure and lesion composition influence drug retention and delivery has evolved. Seminal work by Hwang and Edelman demonstrated that the distribution of hydrophilic dextran in healthy arterial walls is influenced by the arterial ultrastructure, favoring binding to connective tissue elastin and transport along vascular fiber directions. Hydrophobic drugs, such as paclitaxel, which inhibits cell replication through microtubule stabilization, primarily deposit in the intima and adventitia due to specific binding to intracellular tubulin. The success of vascular drug delivery devices depends on tissue binding and clearance [50,51]. Despite their excellent antiproliferative abilities, hydrophilic drugs are less effective in modulating neointimal hyperplasia due to rapid tissue washout [52].

In contrast, hydrophobic drugs, such as paclitaxel and rapamycin, remain resident for days, thanks to their tissue-specific binding capacity. Rapamycin exhibits a more homogeneous distribution than paclitaxel, which favors intimal and adventitial spaces. Clinical practice interchangeably uses different formulations, with recent data supporting sirolimus-based devices in coronary settings. However, challenges persist with sirolimus coatings, such as low lipophilicity and difficulty in release control, which are now addressed through polymer-encapsulated sirolimus coatings for improved long-term efficacy [53,54].

Atherosclerosis and hyperplasia, as pathologic changes, significantly impact drug affinity. Animal models of post-interventional neointimal response have existed, but lesion heterogeneity complicates the study of intraplaque components and drug uptake. Tzafriri et al. evaluated the arterial distribution of paclitaxel, sirolimus, and everolimus in atherosclerotic human and rabbit tissues. Lipid-rich arteries exhibited up to a three-fold lower affinity for hydrophobic drugs due to displaced intracellular binding targets in the lipid-rich environment. Experimental and clinical evidence supports decreased drug deposition with lipids [55,56,57].

Interventional procedures also affect drug uptake and retention. Healthy arteries’ endothelium is a control barrier that angioplasty or stent implantation disrupts. Loss of endothelium facilitates drug diffusion into the tissue, increasing penetration and transmural drug effects [58]. For hydrophobic drugs, such as paclitaxel and rapamycin, endothelial denudation modulates access to drug-specific binding sites, impacting their critical characteristics. Excessive injury during intervention has been suggested as detrimental to drug-eluting stents’ (DES) performance, although further clinical work is needed to verify this mechanism [59].

In atherosclerotic vessels, lesion complexity further complicates drug uptake, leading to contradictory results when lesion morphology is not explicitly assessed. For drug-coated balloons (DCBs), Fernández-Parra et al. observed a four-fold increase in paclitaxel uptake in diseased rabbit vessels, while others reported a decreased uptake. Changes in diffusion, tissue-binding site access, and tissue layer compression may contribute to the observed variations. Thrombus formation around an implanted stent can promote and inhibit drug uptake, depending on its location relative to the stent struts [60,61].

Vascular calcification, prevalent in peripheral arteries, significantly impacts drug permeability by creating an impenetrable structural diffusion barrier. Studies show that preventing vascular calcification by calcium removal increases the drug diffusivity and absorption rate, with clinical evidence linking decreased treatment efficacy to increasing calcium [62]. The morphology of calcified deposits may also influence interventional outcomes, with superficial or sheet-like calcium inhibiting drug uptake more than deep or regionally nodular calcium. However, further experimental validation is needed [63]. In Table 3, some factors implied in the impact of vascular and atherosclerotic structures on drug retention are summarized. These factors collectively contribute to the complex interplay between vascular and atherosclerotic structures and drug retention, highlighting the importance of considering plaque characteristics in drug delivery strategies for the treatment of atherosclerosis.

Hyperplastic vascular response and atherosclerosis have a significant impact on drug delivery. Further, some methodologies used in endovascular drug delivery are presented below. Firstly, the use of bioresorbable stents will be discussed.

### 1.4. Bioresorbable Stents

Bioresorbable stents are designed to dissolve and be absorbed by the body gradually over time. They are typically made from materials such as polylactic acid or magnesium alloy. When implanted in a narrowed or blocked blood vessel, bioresorbable stents provide temporary support to keep the artery open, similar to traditional stents. Over time, the stent material gradually breaks into biocompatible byproducts, allowing the artery to return to its natural state without a permanent implant. This gradual dissolution reduces the risk of long-term complications associated with permanent stents, such as late stent thrombosis or vessel re-narrowing.

Additionally, as the stent dissolves, it allows the artery to regain its ability to constrict and dilate in response to changes in blood flow, which may promote better long-term vascular health. Bioresorbable stents, designed to offer temporary vessel support and fully biodegrade, may or may not release drugs. The potential advantages include a reduced inflammatory response, lower restenosis rates, and the ability to maintain normal vasomotor tone. Unlike permanent metal stents, bioresorbable ones allow for easier reinterventions, dilation beyond the original size, and use in surgical bypasses. They are suitable for pediatric patients and those with metal allergies. Constructed from polymers or metallic alloys, early polymer-based stents showed inflammation issues, but poly-L-lactide (PLLA) stents demonstrated minimal inflammation and durability in porcine models.

Pairing a tyrosine kinase inhibitor with a biodegradable PLLA stent demonstrated a decrease in restenosis. PLLA undergoes degradation over a span of 2 years, with reduced support at 6 months, loss of mass at 12 months, and complete absorption at 24 months [38,54,76].

Bioresorbable alloy stents, primarily crafted from magnesium, have been employed in both animal studies and clinical settings. Magnesium has been chosen because it is an essential mineral well tolerated by the body and absorbed over four months. Metallic bioresorbable stents have specific advantages over polymer-based analogs, including increased strength, rapid degradation, complete radio-opacity, and metal alloys producing only a minimal inflammatory response. However, restenosis rates at four months were identical for the magnesium alloy and the BMS (approximately 38% for both). Although this stent failed to show superiority over BMS in this limited trial, the biocompatibility of the alloy in humans was demonstrated [9,77].

Thus, clinical evaluation of bioresorbable stent technology is ongoing. However, more extensive randomized control trials are necessary to honestly assess the efficacy. Further investigation is needed to verify that this technology is indeed an advance over drug-eluting stents or balloons [78].

Drug-eluting balloons (DEBs), initially developed in the 1980s, are experiencing renewed interest as an alternative to address the limitations of drug-eluting stents (DES). The primary limitation of DES lies in inconsistent drug delivery, with only 15% of the vessel wall in contact with the stent surface, hindering uniform drug elution. DEBs, employing an angioplasty balloon coated with a polymer-eluting antiproliferative agent, provide a solution by enabling uniform and homogeneous coating of the lesion surface. This contrasts with DES, which require longer lengths to cover the entire diseased vessel and are associated with issues such as intimal hyperplasia, lack of adaptive remodeling, and high costs [8,57,79].

Clinical trials actively explore DEBs, mainly utilizing paclitaxel for its efficacy in inhibiting smooth muscle proliferation. Recent advancements in balloon technology, using polymers that enhance drug loading and controlled release, have improved drug retention at the injury site. Clinical trials, such as THUNDER and FemPac, demonstrated positive outcomes, showcasing reduced late lumen loss (LLL) and target lesion revascularization (TLR) rates. LEVANT I and PACIFIER further supported the effectiveness of DEBs in treating femoropopliteal disease, with low LLL and TLR rates reported in the studies. These advancements highlight drug-eluting balloons (DEBs) as a promising solution, particularly in the treatment of infrainguinal diseases [80,81].

Numerous studies are currently underway. Initial data from a multi-center Italian registry revealed high patency rates for the DEB IN.PACT^®^ Amphirion study (Medtronic; Minneapolis, MN) and improved clinical outcomes at one year compared to historical controls of primary angioplasty. Medtronic has made significant investments in its IN.PACT DEB and has initiated several single- and multi-center trials worldwide. The IMPACT SFA I and II trials will assess the efficacy of the IN.PACT^®^ Admiral DEB for femoropopliteal lesions. Trials such as DEBATE-BTK, INPACT DEEP, and PICCOLO are all comparing the IN.PACT^®^ Amphirion paclitaxel-eluting balloon to a non-eluting balloon for below-the-knee vascular lesions. These trials will play a crucial role in determining the effectiveness of DEBs for distal lesions.

The DEBELLUM trial, which is currently underway, will evaluate the same DEB, IN.PACT^®^ Amphirion, for treatment of all infrainguinal diseases [15,82].

In Table 4, some bioresorbable stents are listed. These bioresorbable stents represent advancements in interventional cardiology, offering a temporary scaffold for vessel support while minimizing long-term complications associated with permanent metallic stents, and show the ongoing innovation in the field of interventional cardiology, with a focus on improving deliverability, scaffolding properties, and long-term clinical outcomes.

Another way of delivering drugs is the porous and microporous balloons, which will be presented further in the next section.

### 1.5. Porous and Microporous Balloons

Porous and microporous balloons are used to open narrowed or blocked blood vessels in angioplasty procedures. When inflated, they exert radial force against the vessel walls, which helps compress the plaque and widen the artery. The pores in these balloons allow for the delivery of drugs or contrast agents directly to the vessel wall during inflation. This localized drug delivery can help prevent restenosis or other complications by targeting specific areas of the vessel where the balloon is in contact. The microporous structure may also enhance drug absorption and distribution within the vessel wall, potentially improving treatment outcomes. Overall, porous and microporous balloons combine mechanical dilation with targeted drug delivery to optimize the effectiveness of angioplasty procedures. As balloon technology advances, a variation of drug-eluting balloons, called porous balloons, has emerged. Unlike conventional drug-eluting balloons that coat the balloon’s surface, porous balloons contain the drug within and release it through pores upon inflation. This approach allows for complete and uniform drug delivery along the vessel. Two methods are reported: initial angioplasty with a conventional balloon followed by drug infusion using a porous balloon, and simultaneous drug infusion and angioplasty with the same balloon. Concerns include potential barotrauma-induced intimal hyperplasia, non-homogeneous drug delivery through pores, and systemic drug escape without proper balloon apposition. To address barotrauma, a microporous balloon features an inner porous layer surrounded by an outer membrane with narrower fenestrations to reduce fluid force upon release [92,93,94]. Although reported in a few cases, clinical use of porous balloons awaits validation through large prospective trials. Atrium Medical’s Vascular Clearway irrigation balloon, designed for thrombolytic agent delivery, has shown promise in treating stent restenosis, with successful drug delivery and angiographic evidence of no restenosis in two patients. The ongoing IRRITAX trial sponsored by Atrium aims to evaluate this porous balloon in combination with paclitaxel for treating superficial femoral artery stenosis. Full validation will require significant, randomized trials to assess long-term outcomes [95,96]. Table 5 shows some porous and microporous balloons. These examples demonstrate the use of porous and microporous balloons in various interventional procedures, highlighting their role in drug delivery, embolic protection, and plaque modification.

After discussing some issues regarding drug delivery and methods of endovascular drug delivery, the mechanism of controlling drug release is also essential, and it will be further discussed below.

### 1.6. Mechanisms Controlling Local Drug Release

Controlling drug dose and release kinetics is crucial for effective endovascular drug delivery. Coating drugs onto device surfaces may lead to burst release, potentially overdosing tissues. For controlled release, drug-eluting stents (DES) employ physical mechanisms, such as diffusion, dissolution, and hydrolytic degradation. Chemical mechanisms involve breaking covalent bonds, but their underutilization is due to the need for drug modification. Drug-coated balloons (DCBs) are an alternative to DES, particularly for in-stent restenosis (ISR) and small lesions. DCBs offer a brief drug delivery window, contrasting with DES’s sustained release. Current DES mainly use litmus compounds, while DCBs initially eluted paclitaxel but are exploring different litmus formulations. The impact of physicochemical properties on DCB treatment success is an ongoing investigation [107,108].

Mathematical and computational modeling is a potent tool for simulating drug release and transport in biological environments, particularly in drug-eluting stents (DES). To explain the outcomes, these models consider physiological forces, arterial structures, and drug properties. Early studies revealed the impact of lipophilic vs. hydrophilic compounds, and recent work emphasizes the importance of nonlinear drug-binding phases. Advanced DES kinetics models now incorporate multiple arterial wall layers in realistic 2D-axisymmetric models, providing a more comprehensive understanding [109,110].

While various models exist for drug-eluting stents (DES), there are only a few for drug-coated balloons (DCBs), each with unique strengths and limitations. Commonalities include diffusion-driven drug transport in the arterial wall and the importance of drug binding. However, the models differ in complexity, treatment of other transport processes, dimensionality (1D to 3D), and inclusion of disease factors. Notably, the only model considering the multilayer nature of the arterial wall has limitations in dimensionality and linearity. DCBs and DES exhibit distinct drug delivery kinetics, with DCBs requiring fast delivery and DES allowing sustained release. The optimal drug and release profile to counteract restenosis remains uncertain, and computational models aid in investigating spatiotemporal drug concentrations and comparing safety and efficacy indices. This study simulates drug delivery from DCBs and DES, encompassing different drugs, doses, and release kinetics. Key findings and their potential consequences are reiterated [111,112].

The rapid decline in DC values for the high-dose SIR DCB, along with high levels, suggests that the initial drug loading might be too high. Lowering the initial dose could maintain efficacy while addressing concerns about the high initial DC peak. Conversely, the slower decline of DC for the low-dose SIR DCB, coupled with suboptimal levels, suggests that increasing the initial drug loading could enhance efficacy, possibly with DC peaks within safety margins. A comparison with experimental data showed alignment with existing literature, providing confidence in the simulations [10,113,114]. In the table below, some mechanisms involved in endovascular local drug release are shown. These mechanisms can be employed individually or in combination and offer diverse strategies for achieving precise control over drug release in endovascular applications, catering to specific therapeutic needs and environmental conditions (Table 6).

Furthermore, some strategies implying drug delivery devices are presented.

### 1.7. Device-Based Endovascular Drug Delivery Strategies

#### 1.7.1. Durable Adherent Coatings

Durable adherent coatings are used in medical devices, such as stents, to improve their performance and longevity. These coatings tightly adhere to the surface of the device and are designed to withstand the harsh environment within the body. They typically consist of biocompatible materials resistant to degradation and provide a smooth surface to reduce the risk of clot formation or tissue irritation. Durable adherent coatings help enhance the biocompatibility of medical devices, promote better tissue integration, and minimize adverse reactions within the body, ultimately improving patient outcomes. First- and second-generation DES utilize durable polymer coatings for controlled drug release, falling into two categories: matrix and reservoir types. Similar to the Taxus and Endeavor stents, matrix devices release the drug directly from the polymer matrix into the environment, often termed monolithic devices. Reservoir devices, exemplified by the Cypher and Xience stents, employ at least two layers—an internal drug reservoir and a thin external polymer layer—to regulate drug elution. Hybrid multilayer coating designs offer enhanced control over drug release, optimizing both short-term and long-term biocompatibility [125,126].

In theory, drug release from durable coated drug-eluting stent (DES) designs can be precisely regulated based on the intended thicknesses of the polymer and drug layers, utilizing principles of diffusion and dissolution. However, in practice, multiple rounds of spray coating can lead to significant mixing between consecutive layers and the accumulation of drugs near the surface. For example, while the Cypher stent was initially designed as a reservoir-type DES, imaging studies have revealed drug presence in the outer polymer layer that controls release rates, resulting in a monolithic type of diffusion-controlled release. Since spray coating is traditionally applied to spinning stents, this technique restricts spatial control over the drug coating design [127]. Consequently, optimizing drug delivery between stent struts through differential coating enrichment or applying different drugs to the outer and inner surfaces of stents for restenosis prevention and endothelial promotion poses challenges. Advanced inkjet and micro-drop injection technologies offer improved spatial coating design control at both micro and macro levels. Additionally, the layer-by-layer assembly of polyelectrolytes shows promise for precise device surface engineering and controlled release of charged biological agents, such as DNA and small-interfering RNA [128,129].

Notably, applying spray iterations to static flat metal sheets can overcome the limitations of applying spray to spinning stents. This approach allows for shorter spray run times but longer drying times, reducing mixing between consecutive layers. Furthermore, spraying onto a static sheet in a face-up position is well suited for precise abluminal coating and can be programmed to produce controlled heterogeneous coatings. Recent studies have demonstrated that spraying flat cobalt-chromium sheets with tightly controlled concentrations of the drug Ridaforolimus and two types of polymers resulted in both monolithic and hybrid reservoir/monolithic-type coatings. Drug release from these DES was accurately predicted by a diffusion-based model that considers predetermined layer thicknesses and compositions and a composition-dependent diffusion coefficient [13,130].

Biodegradable-adherent coatings are presented in the next section.

#### 1.7.2. Biodegradable-Adherent Coatings

Biodegradable-adherent coatings are designed to degrade within the body over time, gradually. These coatings are applied to medical devices, such as stents, to provide temporary protection and promote healing while the device is in place. As the coating breaks down, it releases any incorporated drugs or therapeutic agents that the surrounding tissue gradually absorbs. This controlled release helps prevent complications and promotes the desired therapeutic effect. Once the coating has fully degraded, it is naturally eliminated from the body, leaving behind a fully functional medical device. Biodegradable-adherent coatings offer the benefits of targeted drug delivery and reduce long-term risks associated with permanent coatings, making them valuable in various medical applications. Concerns regarding persistent adverse reactions to durable polymeric coatings prompted the development of more biocompatible polymeric materials or bio-erodible coatings absorbed over stent implantation. Some designs aimed to minimize polymer–tissue contact by integrating polymer/drug formulations into sculpted surface inlays within grooves and holes on the struts using micro-dispensers. These grooves and holes vary in sizes, shapes, and positions relative to the struts and can feature the polymer in monolithic, reservoir, or hybrid reservoir/monolithic configurations. Conversely, others attributed safety concerns with durable coated metallic stents to the presence of the metallic scaffold long after acute responses and drug delivery cease, leading to the use of biodegradable polymeric scaffolds as vascular mechanical supports and drug delivery platforms [131,132].

Despite the availability of numerous biodegradable drug-eluting stents (DES) and scaffold designs, achieving the ideal balance between erosion and drug release remains to be fully defined. Many erodible scaffolds and stents with erodible coatings release their entire drug load before erosion occurs, leading to diffusion-limited release kinetics similar to durable coated DES. However, this also inevitably extends the duration of any adverse polymer effects [133]. The belief that sustained delivery of sirolimus analogs is necessary for inhibiting restenosis has led to the duration of drug release from biodegradable coatings and scaffolds being typically similar to that of first-generation Cypher stents. However, this emphasis on release kinetics as the main effect driver, rather than tissue retention, has limited the choice of DES coating materials. Specifically, natural bio-erodible polymers that degrade quickly during hydrolysis and absorption have been avoided in favor of synthetic polymers that are absorbed over several months, such as polylactic acid, polyglycolic acid, copolymer, or similar variations. These artificial materials may cause local irritation due to the release of acidic degradation products, potentially delaying healing and transiently increasing the risk of adverse reactions [3,134].

Evaluation in pig coronary arteries demonstrated sustained efficacious drug levels comparable to those achieved by slow-eluting, durable, coated, sirolimus-eluting stents, leading to superior efficacy and a more favorable tissue response [34,135]. The computational model predicts that sirolimus saturates >65% up to 8 days post-implantation, and the dissociation of the drug FKBP12 complex linearly tracks with coating absorption. Thus, late coating absorption and drug inhibitory effects decline linearly, suggesting that the former drives the latter, representing a new paradigm in stent-based drug delivery [136].

Next, further aspects regarding devices for drug delivery and mechanisms of controlled release, together with the coatings and excipients for drug release, are discussed to introduce the drug-eluting stents in Section 1.11 and Section 1.12 and similar endovascular devices used for drug delivery.

### 1.8. Devices for Vascular Drug Delivery

Early attempts with drug-eluting balloons paved the way for the prominent use of drug-eluting stents (DES) in cardiovascular interventions. The insights gained from DES application have driven the evolution of contemporary drug-coated balloons (DCBs). This overview traces the development of drug delivery devices, emphasizing the intricate connections between device design, drug delivery strategies, and the vascular environment. The interventional success is closely tied to the complexity of the treated lesion. The subsequent sections delve into the evolution of DES, highlighting key developments and underlining the significance of coating modifications in controlled drug release. The fundamental components of DES encompass a mechanical mesh structure ensuring prolonged vessel patency, coupled with a drug-coated coating regulating drug release into the vascular wall. While many DES exist, advancements have often been steered by coating innovations for controlled drug release and long-term elution. Therefore, a comprehensive understanding of controlled release mechanisms and coating technologies precedes the exploration of specific design variations [14].

### 1.9. Mechanisms of Controlled Drug Release in DES

The mechanisms for controlling drug release in drug-eluting stents (DES) can be broadly classified into physical and chemical categories. Physical control involves various methods, such as drug diffusion through a durable polymer coating, release from a dissolving or eroding coating matrix, drug-specific binding via ion exchange or immobilized tissue antibodies, and utilizing osmotic pressure differences to propel the drug into the vascular wall. These mechanisms play a predominant role in DES, with manufacturing parameters, such as polymer formulation, coating thickness, and exposed stent surfaces, directly influencing the predictability of release rates. On the other hand, chemical release control involves breaking the bonds between the drug and its carrier coating to release the drug. However, this method requires chemical modification of the drug to bind to its carrier, resulting in drug release in a prodrug form, which has not gained significant traction in the field of DES. Several major mechanisms are utilized to control drug release in DES, including polymer matrix, diffusion, erosion, ion exchange, and biodegradable coatings. In many DES, a polymer matrix serves as a carrier for the drug, where the drug is dissolved or dispersed within the polymer, gradually releasing in a controlled manner as the polymer degrades over time. Common polymers used include poly(lactic-co-glycolic acid) (PLGA), polyethylene-co-vinyl acetate (PEVA), and poly(styrene-b-isobutylene-b-styrene) (SIBS). In diffusion-controlled drug release, molecules move through the polymer matrix via diffusion gradients, with factors such as molecular weight, size, and concentration gradient influencing the release rate. The properties of the polymer matrix, such as porosity and tortuosity, can be adjusted to modulate the release rate. Erodible polymer matrices are employed in some DES, which degrade over time in response to physiological conditions (e.g., pH and enzymes) or mechanical stress, releasing the encapsulated drug as the polymer degrades. The erosion rate can be tailored by adjusting the polymer’s chemical composition and molecular weight. Certain DES incorporate ion-exchange mechanisms for drug release, where the drug is bound to charged groups within the polymer matrix. Upon exposure to bodily fluids, ions in the fluid compete with the drug for binding sites on the polymer, leading to drug release. Modifying the ionic properties of the polymer can control the release rate. Additionally, DES may feature biodegradable coatings that encapsulate the stent and drug, gradually degrading over time and releasing the drug into the surrounding tissue, with the composition and thickness of the coating influencing release kinetics. Advanced DES may also include mechanisms responsive to external stimuli, such as temperature, light, or magnetic fields, to trigger drug release. These stimuli-responsive materials undergo conformational changes or degradation in response to specific stimuli, enabling controlled drug release [22,137].

### 1.10. Coatings and Excipients for Controlled Release

Various coating and excipient formulations are hypothesized to influence the resulting tissue response, and comprehensive reviews on durable, biodegradable, or deployable coating technologies are readily accessible. Durable coatings, relying on drug diffusion through monolithic or multilayered coatings, offer precise control but raise concerns about hypersensitivity to long-term foreign polymeric presence. Recent innovations, such as durable amphiphilic polymers, silicon carbide, and antibody-based coatings, aim to mitigate inflammatory responses. Biodegradable copolymers, such as poly-L-lactic acid or poly-L-glutamic acid (PLGA), present an active area of research, balancing degradation and release impacts on therapeutic and toxic effects [138,139].

Deployable coatings, an exploratory class, have fewer clinical data. In this category, an absorbable coating is designed to detach and spread into the neointima through tissue remodeling, envisioning sustained effects as neointimal growth encapsulates more of the coating. However, specific release kinetics and transluminal retention patterns need clarification. Deployable coatings with drug delivery in microcrystalline form can be directly related to drug-coated balloons (DCBs), achieving complete delivery during acute inflation [140,141,142].

Regarding drug-binding coatings, the need for dual-antiplatelet therapy (DAPT) is crucial post-DES implantation, but contraindicated patient groups exist. Absorbable coatings might allow a shorter DAPT period, and polymer-free stents have been proposed as an alternative, eliminating the need for DAPT [143,144].

### 1.11. First-Generation DES

First-generation drug-eluting stents (DES) consist of a metal stent scaffold coated with a polymer containing an antiproliferative drug. When deployed in a narrowed or blocked blood vessel, the stent provides mechanical support while the drug-coated polymer gradually releases the medication into the surrounding tissue. This drug inhibits the growth of smooth muscle cells, reducing the risk of restenosis or re-narrowing of the artery. Over time, the polymer coating may degrade or be absorbed by the body, leaving behind the bare-metal stent. Despite their efficacy in reducing restenosis rates compared to bare-metal stents, first-generation DES are associated with potential long-term complications, such as delayed healing and inflammation due to the presence of durable polymer coatings. In the early days of first-generation drug-eluting stents (DES), metallic struts with thicknesses exceeding 100 μm were coated with durable polymers, releasing drugs through diffusion-controlled mechanisms. Notable examples include the sirolimus-based Cypher and paclitaxel-based Taxus DES, both achieving FDA approval with reduced in-stent restenosis rates. Cypher employed a bi-layered design with drug-embedded polymer and a diffusion-limiting top layer, providing additional release control. With its single-layer design, Taxus still allowed considerable diffusion control by adjusting the polymer content. Despite their initial success, first-generation DES were later associated with an increased risk of late stent thrombosis, prompting further evolution in device design. Ongoing improvements targeted reducing the strut size to minimize vascular injury, adopting more potent drug delivery techniques, and emphasizing dual-antiplatelet therapy (DAPT) [145,146].

### 1.12. Second-Generation and Contemporary DES

Second-generation drug-eluting stents (DES) improve upon first-generation designs by addressing limitations such as delayed healing and inflammation associated with durable polymer coatings. These stents typically feature thinner struts and more biocompatible or bioabsorbable polymer coatings. The drug release mechanism remains similar, with an antiproliferative drug gradually released from the polymer coating to inhibit smooth muscle cell proliferation and reduce the risk of restenosis. Additionally, second-generation DES may incorporate modifications to enhance deliverability, flexibility, and conformability within the vessel, improving procedural outcomes and patient comfort. Second-generation DES aim to provide effective and safer treatment options for patients undergoing coronary or peripheral vascular interventions. Second-generation drug-eluting stents (DES) introduced materials with enhanced radial strength, resulting in thinner designs, such as the cobalt-chromium-based Zotarolimus-eluting Endeavour and Everolimus-eluting Xience V. These stents, approximately 30% thinner than their predecessors, showed associations with increased reendothelialization, accelerated vascular healing, and improved clinical outcomes. Modifications in drug formulation and coating morphology were also implemented, leading to improved release control and sustained effects. Sirolimus analogs, such as everolimus and zotarolimus, exhibited enhanced reendothelialization. Introducing novolimus, umirolimus, and ridaforolimus offered further options with unique characteristics. Coating manipulations, including biodegradable and deployable coatings, provided additional regulation of drug delivery [147,148]. Polymer-free DES, releasing the drug directly from the metallic stent backbone, demonstrated significant early release, reaching up to 90% of the total drug load in the initial days. Despite these advancements, the impact of vascular anatomy and lesion morphology on the performance of newer-generation DES has not been extensively studied. Clinical evaluations suggest reduced adverse events and thrombosis rates with second-generation DES, while considerations of lesion complexity and intraplaque composition remain areas of ongoing exploration [19]. While permanent implants, such as DES, have been discussed above, temporary implants, such as bioresorbable scaffolds, are further presented in the next section.

### 1.13. Bioresorbable Scaffolds

Bioresorbable scaffolds are temporary implants used in coronary artery interventions. Made from biodegradable materials, such as polylactic acid, they provide structural support to keep a narrowed artery open, similar to traditional metal stents. Over time, the scaffold gradually dissolves and is absorbed by the body, allowing the artery to return to its natural state. This gradual absorption reduces the risk of long-term complications associated with permanent stents, such as late stent thrombosis. Additionally, as the scaffold dissolves, the artery can regain its ability to constrict and dilate, potentially improving long-term vascular health. Bioresorbable vascular scaffolds (BVSs) utilize a drug-loaded bioresorbable polymer that completely dissolves over time, aiming to restore the vessel to its native state after intervention. The release mechanism involves a diffusion and dissolution control combination, similar to biodegradable coatings in drug-eluting stents (DES). Pharmacokinetic studies have demonstrated comparable release behavior between BVSs and analogous DES. Challenges in finding the right balance between the rate of resorption and the desired drug release rate persist. Nevertheless, the mechanical advantages of bioresorbable vascular scaffolds (BVSs) are underscored, considering that the absence of a metallic implant enables the vessel to restore its native vasomotive, hemodynamic, and constitutive behavior [149,150].

Early bioresorbable vascular scaffolds (BVSs) faced challenges due to the thicker struts required for mechanical stability, leading to an increased surface area and potential adverse effects from local inflammation during mass erosion. Initial clinical experiences were disappointing, with issues such as premature loss of mechanical support, device shrinkage, fracture, and higher rates of in-stent restenosis observed. The first FDA-approved BVS, the everolimus-eluting Absorb stent, was withdrawn from the market due to delayed healing, reduced reendothelialization, and increased rates of restenosis and thrombosis. Revised second- and third-generation BVSs with thinner struts, magnesium-based scaffolds, and improved sizing recommendations exist, but their clinical use remains limited to controlled studies evaluating enhanced designs [151,152].

Bioresorbable vascular scaffolds (BVSs) have encountered challenges and setbacks within a decade of their introduction, with limited comprehensive studies linking them to lesion morphology. Similar to second-generation, biodegradable-coated, drug-eluting stents (DES), BVSs demonstrate hindered drug delivery in lipid-rich or thrombotic environments and reduced effectiveness in calcified lesions. Mechanical stability issues are aggravated by rigid calcified lesions, presenting a specific challenge for BVSs. Initial findings from the Absorb BVS trial suggested the occurrence of neo-atherosclerosis and calcium formation around BVS struts, but substantial conclusions could not be drawn from these early datasets [153]. Figure 1 presents the bioresorbable scaffolds’ mode of action and their advantages compared to metal stents. Additionally, similar to DES, drug-coated balloons are further presented.

### 1.14. Drug-Coated Balloons

Drug-coated balloons (DCBs) are medical devices used in angioplasty procedures to treat narrowed or blocked blood vessels. These balloons are coated with a drug, typically an antiproliferative or anti-inflammatory agent. When the balloon is inflated within the narrowed artery, the drug is transferred directly to the vessel wall, inhibiting smooth muscle cell proliferation and reducing inflammation. This localized drug delivery helps prevent restenosis or re-narrowing of the artery following the procedure. DCBs offer a targeted and practical approach to treating vascular disease, particularly in cases where traditional balloon angioplasty alone may not be sufficient. As an alternative to drug-eluting stents (DES), local drug delivery can be achieved through drug-coated balloons (DCBs). DCBs use a drug-coated angioplasty balloon during vessel reopening, delivering the drug exclusively during inflation and leaving no remaining implant in the vessel. Theoretically, DCBs offer advantages, such as a broader surface area for more homogeneous drug-to-tissue transfer and avoiding delayed arterial healing due to the absence of stent struts. DCBs are considered in scenarios where DES may be impractical, such as in narrow peripheral vessels with high mechanical flexure, where long-term dual-antiplatelet therapy (DAPT) is contraindicated, or in cases of in-stent restenosis (though DCBs are not yet FDA-approved for coronary in-stent restenosis). However, challenges include restricted delivery time, difficulty controlling tissue-retained dose and residence time, and the need for high initial loading. Clinical study results comparing the benefits of DCBs vs. DES in different settings are inconclusive and, currently, DCBs have FDA approval only for use in peripheral vessels [154].

Unlike drug-eluting stents (DES) that provide long-term drug release from permanent stent struts, drug-coated balloons (DCBs) rely on complete drug transfer during balloon inflation. The delivery mechanism for DCBs is not primarily driven by diffusion or dissolution. However, it involves mechanically forcing the drug or drug-carrying coating into the vascular wall during the acute phase. Experimental evidence supports the importance of mechanical contact forces for sensitive DCB delivery. Studies have shown that increasing DCB inflation pressure enhances compulsory drug transfer. However, mechanical adhesion is limiting, as less than 10% of the DCB coating is typically transferred into the wall during routine inflation, resulting in up to 90% of the administered drug dose being lost into the systemic circulation. While amplified vascular injury may improve DCB retention and efficacy, DCBs are still associated with low acute transfer efficiency. Once transferred, transluminal distribution is governed by diffusion kinetics and tissue-specific binding, similar to DES [155,156].

Various drug-coated balloons (DCBs) are available on the clinical market, often utilizing paclitaxel due to its lipophilic characteristics and beneficial protein binding. However, zotarolimus and sirolimus DCBs also exist. While promising results have been achieved in clinical trials, large-scale outcome data remain limited, especially considering lesion morphology. The preference for limus-based formulations over paclitaxel-based DCBs is yet to be decisively established. Recent concerns over paclitaxel-coated devices in peripheral arteries have surfaced, but conflicting results and rapid developments in the DCB field complicate the assessment [157,158].

DCB performance is influenced by the vascular environment, with calcium identified as a hindrance. Lesions with heterogeneous structures may obstruct acute DCB delivery, emphasizing the importance of lesion morphology. Intraplaque thrombus and tissue compression during balloon expansion might also impact optimal long-term delivery. While promising, long-term follow-up data are still needed to assess the full potential of DCBs, especially in coronary settings [159]. While endovascular drug delivery devices have become more advanced and are frequently used, the possible and ongoing transition to lesion-specific drug delivery will be discussed in the next section.

### 1.15. Future Directions: Transition toward Lesion-Specific Drug Delivery

Developing such lesion-specific strategies requires a comprehensive understanding and integration of vascular biology, drug pharmacokinetics, and device mechanics. As research progresses, personalized medicine approaches that tailor interventions to individual patient and lesion characteristics may become more feasible, optimizing the efficacy and safety of local drug delivery devices in treating vascular lesions. The field continues to evolve, and ongoing research and clinical trials are crucial to further refine our understanding and enhance the performance of these innovative devices [160]. As a response to lesion-specific drug delivery, deployable coatings were introduced and are presented further below.

### 1.16. Deployable Coatings

Drug-deployable coatings are medical coatings applied to devices used in vascular interventions, such as stents or balloons. These coatings contain drugs or therapeutic agents released upon device deployment within the blood vessel. When the device is expanded or inflated, the coating comes into contact with the vessel wall, triggering the release of the incorporated drug. This localized drug delivery helps prevent restenosis or other complications by targeting specific areas of the vessel where the device is in contact. Drug-deployable coatings offer a controlled and targeted approach to delivering medications directly to the treatment site, improving the efficacy of vascular interventions. The MiStent sirolimus-eluting stent employs a unique design with an absorbable PLGA coating intended to spread into the neointima through tissue remodeling forces. The microcrystalline sirolimus particles within this coating act as sustained drug delivery micro-depots, ensuring high tissue concentrations long after the stent has reverted to the bare-metal state. This mechanism allows for dynamic drug delivery beyond the immediate vicinity of the stent struts, and computational modeling has demonstrated the impact of coating migration on drug distribution. The computational model predicts that the spread of the coating to inter-strut regions improves drug delivery to these areas and reduces gradients in drug distribution [161,162].

However, beyond a certain migration distance (~100 µm), there is a noticeable decrease in drug deposition near the struts. Peak-trough levels in the media, 150 µm into the tissue, are predicted to decline near-constant with coating migration. While the more homogeneous drug delivery provided by tissue-deployed coatings may have advantages, the optimal dosing and therapeutic efficacy depend not only on total drug levels but also on the concentration of therapeutically active drug bound to its intracellular target. Computational modeling reveals that the MiStent coating achieves a high degree of FKBP12 saturation, even in cases where the coating remains fully conformal to the stent. Compared to conformal stents, deployed coatings saturate a significantly higher percentage of receptors throughout the neointima [163,164].

These findings have implications for the broader class of endovascular delivery devices that deliver drugs in microcrystalline form, including drug-coated balloons and stents with nano-polymer coatings. While these insights are valuable, further quantitative experimental and computational analyses of such devices are needed to comprehensively understand their behavior and efficacy [165].

Polymer-free coated stents are presented in the next section as an alternative to classical stenting with the drug-filled stents, which offer a practical approach to delivering medications directly to the treatment site.

### 1.17. Polymer-Free Coated Stents

Polymer-free coated stents are used in coronary artery interventions without a polymer coating to deliver drugs. Instead, these stents use a proprietary surface treatment to attach drugs or therapeutic agents to the stent surface directly. When deployed in a narrowed or blocked blood vessel, the stent releases the drugs into the surrounding tissue, helping to inhibit smooth muscle cell proliferation and reduce inflammation. This localized drug delivery helps prevent restenosis or re-narrowing of the artery following the procedure. Polymer-free coated stents offer an alternative approach to drug delivery that may reduce the risk of long-term complications associated with traditional polymer-coated stents. Specific metallic drug-eluting stent designs forego polymer coatings entirely, presenting potential benefits, such as avoiding long-term hypersensitivity and thrombogenicity associated with polymer materials, which require extended dual-antiplatelet therapy. Additionally, concerns regarding coating peeling and cracking are mitigated. First-generation polymer-free stents (PFS) were coated in ethanolic paclitaxel [166].

The drug can attach directly to the stent surface through covalent bonding or crystallization–chemical precipitation. Alternatively, the pure drug or a formulation with nonpolymeric excipients can be microinjected into sculpted surface inlays or slots [167].

Similar to drug-coated balloons, second-generation polymer-free stents (PFS) employ non-diffusive mechanisms, notably slow dissolution of sparsely soluble crystalline drug forms, to compensate for the absence of a polymeric layer’s modulating effects. This approach predominantly focuses on hydrophobic drugs, such as paclitaxel and sirolimus analogs, renowned for their established anti-restenotic effects and favorable tissue distribution and retention properties [168]. Clinical experience with second-generation polymer-free stents (PFS) has shown promise, with multiple coronary PFS receiving the CE mark and paclitaxel-eluting peripheral PFS receiving FDA approval. Due to the differences in the eluted drugs, stent geometries, strut thicknesses, surface morphologies, and the paucity of in vivo drug release data, the optimal kinetics of drug elution from PFS have yet to be defined [169].

The efficacy of sirolimus analog release kinetics in paclitaxel-filled stent devices can be understood by examining tissue delivery profiles and contrasting them with bioresorbable-coated and durable-coated sirolimus drug-eluting stents (DES) [170].

For instance, the Yukon PFS releases 66.4% and 85.5% of its sirolimus load within 7 and 21 days of in vitro deployment, respectively. Tissue concentrations during the first week of implantation surpass those provided by slow-eluting Cypher stents but decline significantly by day 10. The Yukon Choice stent, incorporating a biodegradable polymer with sirolimus, releases the drug at a nearly constant rate during 28-day deployments, sustaining high tissue levels up to 20 days post-implantation. Clinical late lumen loss in de novo coronary lesions correlates with the duration of tissue retention, as Yukon PFS, but not Yukon Choice, is inferior to durable-coated Cypher stents [171,172].

The predictive power of local tissue concentrations over the released dose becomes evident for the Biofreedom PFS, which removes the highly lipophilic sirolimus analog Biolimus A9. Although Biofreedom PFS releases ~90% and ~99.9% by 2 and 28 days, the associated tissue concentration in porcine arteries at 28 days is comparable to that sustained by slower-eluting durable-coated DES. This sustained tissue concentration reduces inflammation and wall thickening compared to slower-eluting, durable-coated Cypher stents at 28 and 180 days. Clinical studies also show non-inferior late lumen loss, significantly lower significant cardiac event rates, and target lesion revascularization for Biofreedom PFS compared to Taxus stents at two years [173].

The extreme lipophilicity of Biolimus A9 relative to other sirolimus analogs is suggested to contribute to its favorable tissue absorption and cell uptake, highlighting the importance of pharmacokinetic profiles in the efficacy of PFS devices. Further investigation, including animal and computational models, is needed to substantiate these assertions [174].

### 1.18. Drug-Filled Stents

Drug-filled stents are used in vascular interventions that contain a reservoir or coating filled with medication. These stents release the drug directly into the surrounding tissue upon deployment, helping to inhibit smooth muscle cell proliferation and reduce inflammation. This localized drug delivery helps prevent restenosis or re-narrowing of the artery following the procedure. Drug-filled stents offer a practical approach to delivering medications directly to the treatment site, improving the efficacy of vascular interventions while reducing the risk of systemic side effects associated with oral medications. The drug-filled stent (DFS), a new polymer-free drug-eluting stent technology from Medtronic, features tubular stent struts with a hollow core and small access holes for drug release. Initial 90-day pig studies with a prototype sirolimus-eluting DFS showed comparable drug release rates and tissue concentrations to the Resolute stent. It effectively suppressed neointimal hyperplasia at 28 days compared to bare-metal stents, with minimal inflammation observed through 90 days. Biexponential sirolimus release kinetics were noted, suggesting both immediate and sustained release pools modulated by hole size. Ongoing modeling analysis aims to understand this system further [175,176,177].

## 2. Discussion

The field of endovascular drug delivery has experienced a recent resurgence of innovation aimed at improving performance and reducing costs. Fundamental aspects of first-generation drug-eluting stents (DES) have been reevaluated, challenging the need for a persistent metallic scaffold, adherent polymer coatings as drug release reservoirs, and sustained drug release [178]. There are varying opinions, with some advocating for eliminating polymer coatings while others explore new roles, such as using polymers as deployable carriers of crystalline drugs. This review emphasizes the importance of quantitative experiments and computational modeling in understanding the intricate interplay between device design, drug release, tissue distribution, and therapeutic effects. These techniques provide a crucial framework for ongoing and future innovations in the field. Key points highlighted include combination drug-eluting devices, where the effects of combination drug-eluting devices are multifactorial, and tissue drug concentrations’ predictive effect, whereby drug concentrations in tissues predict therapeutic effects, and achieving adequate drug distribution and retention is crucial. These techniques have led to diverse designs and ongoing innovation in this space. Further, promising drug pharmacology is requisite: while favorable drug pharmacology is essential, the focus should extend to achieving optimal drug distribution and tissue retention. Another point is that computational modeling drives innovation, and understanding and computationally modeling drug release kinetics and tissue distribution determinants is vital in driving innovation, potentially at reduced costs. These considerations shape the dynamic landscape of endovascular drug delivery, and ongoing research will likely yield further advancements in the field [179,180,181]. Further, the clinical implications of lesion-specific intervention are discussed.

### 2.1. Clinical Implications for Lesion-Specific Intervention

The anticipated clinical transition toward lesion-specific intervention necessitates a holistic approach that is not reliant on a single device or intervention adjustment. It entails simultaneous modifications in lesion characterization methods, device selection, and drug formulations. Moreover, purposefully designed clinical trials are crucial to validate the significance of lesion-specific characterization in clinical practice [182,183].

Lesion-specific characterization through medical imaging is essential for advancing tailored interventions. Intravascular imaging, particularly virtual histology by intravascular ultrasound (VH-IVUS), allows detailed insights into plaque morphology. VH-IVUS classifies tissue into fibrotic, fibro-fatty, calcific, and necrotic core categories, validated for coronary lesions. It also correlates the lesion phenotype with hyperplastic development in coronary drug-eluting stents (DES). Near-infrared spectroscopy enhances lipid detection alongside VH-IVUS. Optical coherence tomography provides high-resolution imaging, aiding vascular drug delivery and identifying calcific developments affecting delivery. Recently, optical coherence tomography has incorporated virtual histology-like features. It is desirable to develop similar micromorphology characterization from noninvasive angiography, considering its widespread use in clinical practice, albeit with existing qualitative scores on calcium burden [184,185,186].

Imaging is vital in quantifying lesion morphology and is expected to be pivotal in an era focused on lesion-specific interventions. Numerous examples illustrate the effective integration of imaging into clinical practice for identifying vulnerable plaques, assessing acute and long-term treatment efficacy, and evaluating regional lesion status before drug-eluting stent (DES) implantation. A recent meta-analysis underscored improved patient survival by incorporating intravascular imaging and lesion assessment before DES implantation. In the context of a lesion-specific transition, quantifying lesion morphology through medical imaging could inform the selection of devices and drugs for interventional treatment. For example, lipid-rich plaques might suggest sirolimus-based delivery due to enhanced tissue-specific paclitaxel binding displacement.

Conversely, hemorrhagic or thin fibrotic structures might indicate structural instability, necessitating rigid stent-based delivery [187,188]. To definitively assess the benefits of morphology-resolving imaging, image-derived lesion morphology should be incorporated as a focal point in clinical evaluations of drug delivery devices. This entails monitoring the volumetric content, anatomic positioning (superficial vs. deep), and morphologic regionality (local vs. circumferential) concerning clinical outcomes. [189,190].

Similarly, image-based lesion characterization could assist in selecting appropriate lesion preparation strategies, potentially enhancing subsequent retention of delivered drugs. Recent review literature emphasizes that identifying highly calcific structures through imaging could justify pre-procedural removal or modification to improve drug permeation. Likewise, detecting lipid-rich areas could offer insights into mechanical stability, displacement of critical tissue-binding sites, or even reveal the actual lesion length in complex phenotypes. Overall, lesion preparation could help standardize vascular response and drug retention by adapting to changing morphologic structures, with imaging playing a pivotal role in elucidating definitive vascular mechanisms driving such behavior. While comprehensive clinical data on the impact of lesion preparation on adjunctive drug-eluting technologies is still pending, ongoing trials, such as Disrupt CAD/PAD I–III and ECLIPSE, hold promise for generating such data soon [191,192,193,194].

Lesion-specific intervention extends to the choice of drug and device. Imaging provides insight into morphological characteristics, influencing the selection of intervention strategies. For drug choice, there is ongoing debate in the interventional field. Levin et al. reviewed that transluminal retention relies on drug-specific binding sites, with sirolimus-based drugs targeting mammalian targets of rapamycin receptors distributed more evenly through the vessel wall. In contrast, paclitaxel primarily targets tubulin in the subintimal and adventitial layers. Device-induced intimal disruption or atherosclerotic displacement of binding sites may be more detrimental to paclitaxel-based delivery.

Preclinical studies have demonstrated that sirolimus binding is less sensitive to lesion complexity, whereas paclitaxel-based partition coefficients decline rapidly with increasing lipid content [195,196,197]. Complex lesions with an active core and distinct lipidic components may benefit from sirolimus-based delivery. Long-term retention can be enhanced by prioritizing sirolimus formulations with high partition coefficients, such as the more hydrophobic zotarolimus or everolimus compared to sirolimus. The preference for paclitaxel delivery remains in cases of high medial-to-deep calcification, where profound permeation barriers shift binding to subintimal spaces. Assuming minimal subintimal disruption from the implanted or inflated device, the higher partition coefficient of paclitaxel could lead to a more efficient antiproliferative effect. In contemporary practice, paclitaxel delivery also appears favored in calcified peripheral lesions [198,199,200]. However, lesion preparation could present a more feasible alternative to address the specific environment before intervention in such scenarios. Experimental and clinical validation of paclitaxel vs. sirolimus in such contexts is still pending. A shift toward lesion-specific approaches requires conclusive clinical evidence guided by purposefully designed clinical trials [55,201].

Besides the influence of lesion complexity on drug formulation, the visible lesion morphology also guides the selection of interventional devices. The difference in delivery mode between drug-coated balloons (DCBs) and drug-eluting stents (DES) or bioresorbable vascular scaffolds (BVSs) is crucial to consider in lesion complexity. DCBs rely on short-term mechanical deposition during inflation, facilitating delivery into soft, hematoma-rich, fatty lesions (e.g., in-stent restenosis tissue), but obstructing delivery into fibro-calcific or calcific lesions. Coating micromorphology may help overcome more rigid intimal entities, with microneedle configurations inducing higher contact pressure and coating transfer than amorphous coating equivalents. In highly calcific lesions, stent-based intervention and appropriate vessel preparation are preferred. Atherectomy reduces superficial calcium through physical grinding, while contemporary lithotripsy-based techniques address deep calcium. Vessel preparation may rectify divergent results stemming from heterogeneous lesion phenotypes, yet conclusive large-scale clinical outcome results in this area are still pending [202,203,204].

Considering the heterogeneity of atherosclerotic lesions, we foresee lesion-specific intervention as transitioning toward tailored vessel preparation techniques and delivery devices guided by lesion characteristics rather than adhering to a one-size-fits-all approach [180,205].

The shift toward lesion-specific drug delivery intervention benefits from continuous updates and refinements driven by advanced computational and scientific evaluations. High-fidelity imaging enables patient-specific modeling for predictive assessments of intervention outcomes. Computational simulations evaluate factors governing hemodynamic, structural, and drug responses. Integration of pharmacokinetic and pharmacodynamic profiles enhances understanding of drug delivery mechanisms. Recent advancements aim to include lesion-specific components in simulations, highlighting the ongoing role of computational modeling in establishing guidelines [206,207,208].

Nonhuman animal modeling is crucial for establishing frameworks for lesion-specific intervention. Animal models offer a controlled study environment, allowing isolated evaluations while retaining the complexity of the cardiovascular system. They have been central to developing local drug delivery devices, considering lesion-specific aspects. Understanding that animal and computational modeling should not precede clinical evaluations is essential. Instead, developing local drug delivery devices should involve iterative evaluations encompassing theoretical, computational, animal, and clinical assessments [209,210]. The following section presents the future of lesion-specific interventions and endovascular drug delivery perspectives.

### 2.2. Lesion-Specific Intervention and Future Drug Delivery: Looking toward the Horizon

The transition toward lesion-specific interventions and improved clinical outcomes does not necessarily require novel coating designs, new drug formulations, or updated device specifications. Instead, the performance of existing devices could be enhanced by recognizing the morphologic scenarios in which they are most effective. High-fidelity imaging techniques can play a crucial role in achieving such morphologic assessments. Promising trends in new device designs involve novel materials, such as tantalum- or nitinol-based alloys, which offer flexible yet durable low-profile struts, promoting early reendothelialization and reducing vascular injury. Sirolimus-based formulations are favored in contemporary results, especially in complex phenotypes, and recent developments in drug-coated balloons (DCBs), using polymeric capsules within deployable coatings, show promise for sirolimus-based delivery [83,211]. Considering lesion-specific treatment, vessel preparation becomes an essential aspect of future drug delivery procedures to achieve homogenization of the vessel response. Molecular biology and genetics advances hold promise for cell-selective drug formulations or gene-eluting stents. However, translating these technologies into clinical implementation and understanding their role in lesion-specific intervention requires further work. Incorporating any of these techniques must be closely coupled with assessing lesion morphology in future clinical and nonclinical studies, recognizing its governing influence on local drug delivery, vascular behavior, and interventional outcomes [78,212]. In the table below, some future directions are listed. These future directions hold promise for advancing endovascular drug delivery, offering innovative approaches to address current challenges and improve clinical outcomes. Future directions represent ongoing research efforts aimed at improving the efficacy, safety, and precision of endovascular drug delivery for various vascular diseases (Table 7).

The figure below shows a schematic diagram depicting the process of endovascular drug absorption (Figure 2).

The last sections present external drug delivery devices and a brief description of systemic drug administration with endovascular absorbtion and the target intravenous drug delivery.

### 2.3. Extraluminal Drug Delivery Devices and Perivascular Biomaterials

Extraluminal drug delivery devices and perivascular biomaterials are designed to deliver medication around blood vessels rather than directly into them. These devices typically involve the placement of drug-eluting implants or coatings around the outside of the blood vessel, targeting the tissue surrounding the vessel rather than the vessel lumen itself. The medication is gradually released from these devices, diffusing into the surrounding tissue to inhibit smooth muscle cell proliferation, reduce inflammation, or achieve other therapeutic effects. This approach can be beneficial in cases where direct intraluminal drug delivery is not feasible or practical, providing a targeted and localized treatment option for vascular diseases. Periadventitial drug delivery via a perivascular wrap offers an alternative approach. In a study by Kelly et al., ethylene vinyl acetate perivascular wraps loaded with paclitaxel were evaluated in a porcine model of arteriovenous graft stenosis. Anastomoses treated with paclitaxel-loaded wraps during surgery showed reduced luminal stenosis compared to untreated graft-vein anastomoses (0.17% in the paclitaxel group vs. 37.90% stenosis in the control group) [165,224].

Poly(ε-caprolactone) (PCL), a biocompatible and biodegradable polymer, has been utilized for delivering paclitaxel and rapamycin to the vessel wall. Extensive in vitro and in vivo studies have led to FDA approval for various medical drug delivery devices composed of PCL. In a mouse femoral artery injury model, PCL cuffs loaded with paclitaxel or rapamycin and control cuffs were placed around injured femoral arteries. After three weeks, paclitaxel and rapamycin-eluting PCL cuffs reduced intimal thickening by 76% and 75%, respectively, with localized delivery and no observed adverse systemic effects. [225].

Although showing promising results in animal models, the transition of this technology to human clinical trials has been sluggish. Only one company, VesselTek Biomedical (Chicago, IL, USA), is currently nearing human trials. They are developing a drug-eluting perivascular wrap (VTek-RA wrap) made from poly(diol citrate) [226].

Injection catheters are intraluminal devices equipped with microneedles, designed to deliver drugs to the periadventitial space. Upon insufflation, these microneedles pierce the vessel wall, releasing drugs into and around the vessel’s adventitia. This method allows drug delivery to the outer vascular wall through a percutaneous intervention, minimizing the direct impact on the vessel lumen and reendothelialization. Studies have investigated the effectiveness of injection catheters in delivering drugs to the porcine femoral artery, such as paclitaxel and nab-rapamycin, demonstrating reductions in neointimal area and restenosis [227].

However, injection catheters have some drawbacks, including the potential for uneven drug distribution and concerns about toxicity if excessive doses are administered to a localized area. Mercator MedSystems has introduced a micro-infusion catheter with a single injection needle, which has received FDA 510(k) clearance. Although there is limited published literature on this device, preliminary data presented at the Society for Vascular Surgery Annual Conference suggests its readiness for clinical safety trials. Additionally, injection catheters developed by Bavaria Medizin Technologie GmbH (Wessling, Germany) and Binlab, Inc. (Webster, TX, USA) have been patented, but no published data currently describe their efficacy [228]. Some perivascular devices are listed in Table 8. These additional examples highlight the diversity of extraluminal (perivascular) drug delivery devices, offering targeted treatment options for various vascular diseases and conditions.

### 2.4. Systemic Drug Administration

Intravenous drug delivery has been a common approach for delivering drugs systemically. Paclitaxel, initially approved in December 1992 under the trade name Taxol, is administered intravenously. However, the conventional form of intravenous paclitaxel contains ethanol and Cremophor^®^ EL, leading to severe hypersensitivity reactions. Patients must be premedicated with corticosteroids and antihistamines before treatment [238].

In January 2005, a novel albumin-bound nanoparticle version of paclitaxel (nab-paclitaxel or Abraxane^®^) was approved for the treatment of metastatic breast cancer. This modification improved paclitaxel’s efficacy and safety profile by eliminating the need for premedication. Several clinical trials, including SNAPIST-I, II, and III, assessed nab-paclitaxel (Coroxane^®^) to prevent restenosis after stenting bare-metal de novo coronary lesions. However, nab-paclitaxel can lead to side effects, such as neutropenia, leucopenia, and alopecia, particularly at high doses [239].

Other systemically administered pharmacotherapeutic regimens have effectively prevented restenosis in animal models. However, their success in human trials has been limited due to poor tolerance and a narrow therapeutic range for these drugs. Ongoing research aims to explore and develop new systemic approaches to prevent restenosis in a more targeted and effective manner [240].

### 2.5. Targeted Intravenous Drug Delivery

Utilizing tissue-specific targeted systemic treatments for local drug delivery shows promise. In this approach, agents are administered systemically but include tissue-specific tags that guide them to the injured vessel following vascular intervention. Deglau et al. investigated a site-specific delivery system employing microspheres carrying therapeutic drugs. These microspheres consisted of reactive polyethylene glycol tagged with avidin. Using the Remedy microporous balloon, a balloon coated the injured artery with biotin. Avidin-coated microspheres were then intravenously administered, attaching to the biotin on the arterial wall and locally releasing the drug. This innovative approach has the potential to precisely deliver systemically injected anti-restenotic agents to the target site, offering a more targeted and effective treatment strategy [241].

Targeted drug delivery through systemic injection using microspheres represents an innovative method. These microspheres can be engineered to target proteins specifically expressed or upregulated after vascular injury. For example, researchers have developed microspheres or particles that directly target surface markers exposed following vascular injury, such as E- and P-selectin, ICAM-1, and VCAM-1. Similarly, nanoparticles can be modified to target and bind to specific proteins within the injured arterial wall. In a study by Chan et al., they designed a nanoparticle with a lipid core–shell interface between polylactide-co-glycolic acid and polyethylene glycol polymers. This nanoparticle was loaded with paclitaxel and had peptides directed against collagen IV, aiming to bind to collagen IV in the basal lamina of the vessel wall exposed after endothelial denudation from mechanical injury. Safety studies in rats showed no signs of toxicity. However, further clinical trials are necessary to assess the safety and efficacy of these innovative techniques in human subjects [242]. Some examples are listed in the table below (Table 9), where the most used endovascular targeted devices are listed together with their implications in intimal hyperplasia.

## 3. Conclusions

Vascular drug delivery devices, such as drug-eluting stents (DES), bioresorbable vascular scaffolds (BVSs), and drug-coated balloons (DCBs), play a crucial role in cardiovascular medicine. However, the effectiveness of these devices is intricately tied to the specific characteristics of the treated lesions. The emphasis on lesion-specific intervention recognizes the importance of tailoring therapy based on the unique features of the atherosclerotic lesions and vascular environment. Advances in imaging technologies, predictive modeling, and innovative delivery techniques are expected to contribute to this transition, enabling more refined and effective endovascular therapies.

While progress has been made in addressing restenosis in coronary arteries, challenges persist in peripheral vessels. Nevertheless, promising approaches, including drug-eluting balloons, periadventitial drug delivery, and targeted systemic therapies, suggest that solutions for preventing restenosis in peripheral vessels may be on the horizon. The ongoing advancements in the field provide hope for improved outcomes in treating vascular diseases.

Local drug delivery has become a prominent strategy for preventing restenosis after interventions for obstructive atherosclerosis. While there is considerable enthusiasm for this technology, there is a need for a deeper understanding of the therapeutic mechanisms involved. The success of locally administered drugs is closely tied to the vessel’s anatomy and the morphology of the lesion being treated. As our understanding of lesion-specific therapies continues to evolve, there is potential for the expanded clinical application of local drug delivery devices. This approach recognizes the importance of tailoring interventions to the specific characteristics of the atherosclerotic lesions, paving the way for more effective and targeted treatments.

## Figures and Tables

**Figure 1 life-14-00451-f001:**
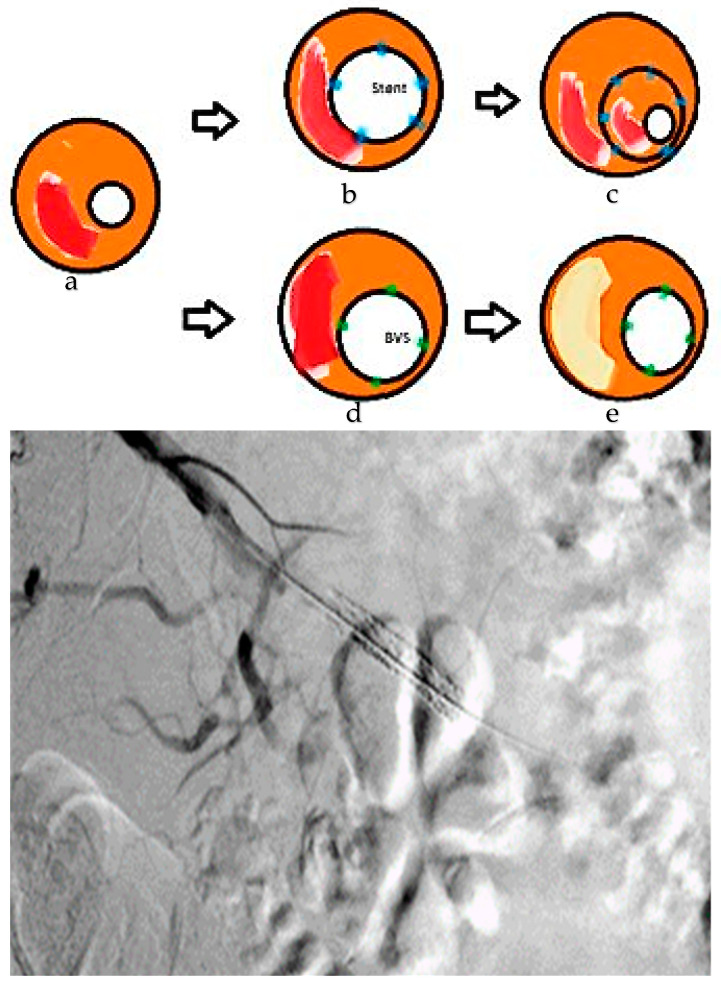
Bioresorbable scaffold’s mode of action. (**a**) Artery with atherosclerotic plaque with significant arterial stenosis (plaque represented in red spot). (**b**) A stent is deployed at the level of the stenosis, increasing the circulating arterial lumen and, consecutively, an increase in flow. (**c**) In time, due to the intimal hyperplasia, the stent is covered with hyperplasic intima and, consecutively, new plaque is formed, obstructing the lumen again, resulting in a more significant narrowing of the artery. (**d**) A bioresorbable scaffold is placed, this time in the obstructed arterial lumen, having the same results as the stent. (**e**) In time, a bioresorbable scaffold creates a film over the atherosclerotic plaque, the endothelium. The artery expands, but the intimal hyperplasia process is at a minimum. The scaffolds are resorbed, leaving a much greater lumen and, consecutively, a greater arterial flow than the stent. In the lower image (personal collection), a stent is represented by mild signs of intimal hyperplasia at the stent level.

**Figure 2 life-14-00451-f002:**
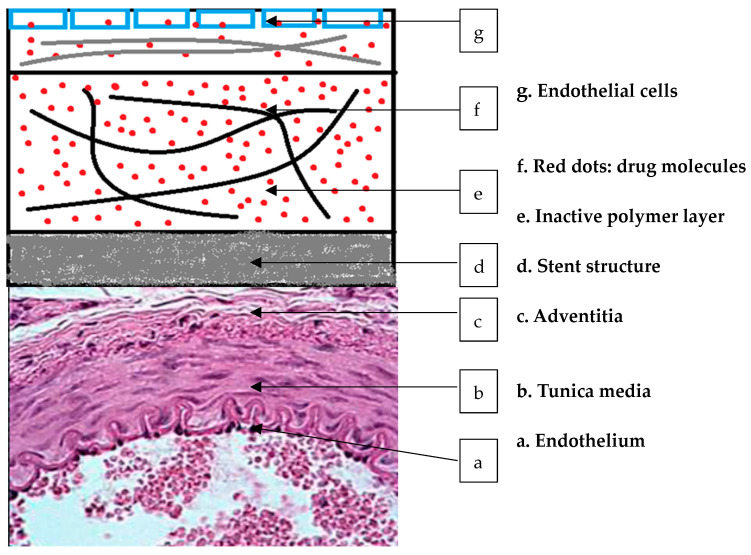
Mechanism of endovascular absorbtion. The figure above considers a stent covered with a polymer drug-containing layer placed in an artery. The stent metal structure is represented in grey (**d**). On top of the stent structure is an inactive polymer layer that cannot release the drug molecules (**e**). Near the endothelium (**g**), a thin layer of active polymer can release the drug’s active molecules-the drug molecules are represented in ref dots (**f**). The drug release molecules interact with the endothelium and perform their pharmacological role. The figure below shows a section through the human artery, stained with hematoxylin–eosin (personal collection). The endothelium is shown (**a**), followed by the tunic media (**b**) and adventitia (**c**). The drug is released at the endothelial layer and absorbed into the tunica media.

**Table 1 life-14-00451-t001:** Intraluminal drug delivery devices.

#	Device	Description	Ref.
1	Drug-eluting stents (DES)	Stents coated with drugs to prevent restenosis (re-narrowing) of the blood vessel after angioplasty.	[22]
2	Intraluminal brachytherapy catheters	Catheters are used to deliver localized radiation therapy within body lumens and are often used in conjunction with other cancer treatments.	[23]
3	Intraluminal drug delivery balloons	Balloon catheters coated with drugs for localized drug delivery during angioplasty procedures.	[24]
4	Drug-coated balloons (DCB)	Balloon catheters coated with drugs to deliver medication directly to the vessel wall during angioplasty.	[25]
5	Intraluminal drug delivery capsules	Capsules are designed to release drugs in the gastrointestinal tract for targeted delivery.	[26]
6	Intraluminal drug-eluting spheres or beads	Microspheres or beads loaded with drugs for targeted delivery within body lumens, such as blood vessels or the gastrointestinal tract	[27]
7	Intraluminal drug-eluting films or coatings	Thin films or coatings are applied to luminal surfaces, releasing drugs over time for localized therapy.	[28]
8	Intraluminal drug delivery nanoparticles	Nanoparticles are designed to deliver drugs to specific sites within body lumens, offering targeted therapy.	[29]
9	Intraluminal drug-eluting sponges or gels	Sponges or gels are impregnated with drugs for controlled release within luminal spaces, such as the urinary tract.	[30]
10	Intraluminal drug delivery microcapsules	Microcapsules containing drugs for controlled release within luminal environments, such as the gastrointestinal tract	[31]

**Table 2 life-14-00451-t002:** Factors implicated in hyperplastic vascular response.

#	Factor	Description	Ref.
1	Inflammatory response	Inflammatory processes are crucial in developing a hyperplastic vascular response, leading to smooth muscle cell proliferation and migration.	[38]
2	Smooth muscle cell proliferation and migration	Hyperplasia involves the excessive proliferation and migration of smooth muscle cells, contributing to vessel lumen narrowing.	[39]
3	Extracellular matrix remodeling	Changes in the extracellular matrix composition and remodeling contribute to the hyperplastic response by providing a scaffold for smooth muscle cell proliferation.	[40]
4	Platelet activation and thrombosis	Platelet activation and subsequent thrombus formation can trigger the hyperplastic vascular response by initiating inflammatory cascades and smooth muscle cell activation.	[41]
5	Neointimal formation	Neointimal formation, characterized by the proliferation of smooth muscle cells and extracellular matrix deposition, is a hallmark of hyperplastic vascular response.	[42]
6	Intracellular signaling pathways	Various intracellular signaling pathways regulate smooth muscle cell proliferation and migration, including growth factors, cytokines, and mitogen-activated protein kinases (MAPKs).	[43]
7	Endothelial dysfunction	Dysfunction of the endothelial layer of blood vessels can lead to impaired vasodilation, increased inflammation, and enhanced smooth muscle cell proliferation, contributing to restenosis.	[44]
8	Matrix metalloproteinases (MMPs)	MMPs are involved in extracellular matrix degradation and remodeling, which play a role in vascular remodeling and restenosis.	[45]
9	Oxidative stress	Excessive production of reactive oxygen species (ROS) leads to oxidative stress, which promotes smooth muscle cell proliferation, inflammation, and vascular remodeling.	[46]
10	Hypoxia-inducible factor (HIF) pathway	Hypoxia-inducible factors are transcription factors that regulate cellular responses to hypoxia. Activation of the HIF pathway can promote neointimal formation and restenosis.	[47]
11	Vascular endothelial growth factor (VEGF)	VEGF is a potent angiogenic factor involved in neovascularization and vascular remodeling. Dysregulation of VEGF signaling can contribute to restenosis.	[48]
12	Genetic factors	Genetic variations in genes involved in inflammation, smooth muscle cell proliferation, and extracellular matrix remodeling can influence susceptibility to restenosis.	[49]

**Table 3 life-14-00451-t003:** Vascular and atherosclerotic structures’ impacts on drug retention.

#	Factor	Description	Ref.
1	Endothelial permeability	The permeability of the endothelial layer can affect drug penetration into the vascular wall and atherosclerotic plaques.	[64]
2	Plaque composition	The composition of atherosclerotic plaques, including lipid content, fibrous cap thickness, and calcification, can influence drug retention and distribution within the plaque.	[65]
3	Vascular architecture	The structure and geometry of blood vessels, including vessel diameter, branching patterns, and tortuosity, can affect drug distribution and retention.	[66]
4	Macrophage infiltration	Macrophages play a crucial role in atherosclerosis and can serve as drug targets. Drug retention may be influenced by the density and activity of macrophages within the plaque.	[67]
5	Neovascularization	Neovascularization within atherosclerotic plaques can affect drug delivery and retention. New blood vessels may enhance drug penetration into the plaque or provide routes for drug escape.	[68]
6	Calcification	Calcium deposits within atherosclerotic plaques can affect drug distribution and retention by altering the local microenvironment.	[69]
7	Lipid core size and stability	The size and stability of the lipid core within atherosclerotic plaques can influence drug retention and release kinetics.	[70]
8	Plaque erosion and rupture susceptibility	Plaque erosion and rupture susceptibility can affect drug retention by altering atherosclerotic plaque’s surface properties and integrity.	[71]
9	Collagen content and stability	Collagen content and stability within the fibrous cap of atherosclerotic plaques can impact drug retention and release kinetics.	[72]
10	Hemodynamic forces	Hemodynamic forces, such as shear stress and turbulence, can affect drug distribution and retention within the vasculature and atherosclerotic plaques.	[73]
11	Local pH and electrostatic interactions	Local pH and electrostatic interactions within the plaque microenvironment can influence drug retention and release kinetics.	[74]
12	Matrix metalloproteinases (MMPs) activity	MMP activity within atherosclerotic plaques can affect drug retention by altering the extracellular matrix structure and composition.	[75]

**Table 4 life-14-00451-t004:** Bioresorbable stents.

#	Stent	Description	Ref.
1	Absorb Bioresorbable Vascular Scaffold (BVS; Abbott Vascular, Abbott Park, IL, USA)	The Absorb BVS is a polymeric stent made of poly-L-lactic acid (PLLA) coated with a bioresorbable polymer that elutes everolimus, an antiproliferative drug	[83]
2	Magmaris Magnesium Bioresorbable Scaffold (Biotronik, Bulach, Switzerland)	The Magmaris scaffold is a magnesium alloy coated with a bioresorbable polymer containing sirolimus, an antiproliferative drug.	[84]
3	DESolve Bioresorbable Coronary Scaffold System (Elixir Medical Corporation, Milpitas, CA, USA)	The DESolve scaffold comprises a poly-L-lactic acid (PLLA) polymer and elutes novolimus, a limus-based drug.	[85]
4	REVA Medical Fantom Bioresorbable Scaffold, (REVA, San Diego, CA, USA)	The Fantom scaffold is made of a tyrosine-derived polymer and coated with a bioresorbable polymer containing sirolimus.	[86]
5	ART Pure Bioresorbable Stent (Arterial Remodeling Technologies, Paris, France)	The ART Pure stent comprises a PLLA scaffold without a polymer coating and is designed to provide support during healing before complete resorption.	[87]
6	Synergy Bioabsorbable Polymer Stent (Boston Scientific, Marlborough, MA, USA)	The Synergy stent is a cobalt-chromium platform coated with a bioresorbable polymer containing everolimus, designed to minimize polymer exposure after drug elution.	[88]
7	Fantom Encore Bioresorbable Scaffold (REVA Medical, San Diego, CA, USA)	The Fantom Encore scaffold is an improved version of the Fantom scaffold, offering enhanced visibility under fluoroscopy and a modified strut design for improved deliverability.	[89]
8	MeRes100 Bioresorbable Scaffold (Meril Life Sciences, Gujarat, India)	The MeRes100 scaffold is a PLLA-based stent with a thin-strut design and a hybrid coating of a bioresorbable polymer containing sirolimus.	[90]
9	DESolve Nx Bioresorbable Coronary Scaffold (Elixir Medical Corporation, Milpitas, CA, USA)	The DESolve Nx scaffold is an updated version of the DESolve scaffold with thinner struts and a novel polymeric matrix designed to improve deliverability and vessel scaffolding.	[91]
10	RT DIVA Bioresorbable Stent (Arterial Remodeling Technologies, Paris, France)	The ART DIVA stent is a PLLA-based scaffold designed for use in small vessels, offering radial strength and support while minimizing the risk of restenosis.	[24]

**Table 5 life-14-00451-t005:** Porous and microporous balloons.

#	Balloon	Description	Ref.
1	DuraPleat^®^ Balloon (Convergent Therapeutics, Boston, MA, USA)	The DuraPleat balloon is a porous balloon designed for drug delivery applications, featuring a unique pleated design to increase the surface area and drug elution capacity.	[97]
2	PercuSurge GuardWire^®^ Plus Temporary Occlusion and Aspiration System (Medtronic, Dublin, Ireland).	The PercuSurge GuardWire Plus system includes a microporous balloon for distal embolic protection during angioplasty and stent placement procedures.	[98]
3	Innovative Angioplasty Technologies SonicBlast^®^ Micro-Porous Balloon (Innovative Angioplasty Technologies, Michigan, MI, USA)	The SonicBlast Micro-Porous balloon features micropores along its surface, allowing for improved drug delivery and enhanced vessel wall penetration.	[99]
4	Pantheris Atherectomy System (Avinger, Redwood City CA, USA)	The Pantheris system includes a porous balloon for vessel occlusion and blood flow management during atherectomy procedures, facilitating plaque removal in peripheral arterial disease.	[100]
5	Istent Atherectomy System (Ra Medical Systems, Carlsbad, CA, USA)	The Istent Atherectomy System includes a microporous balloon for plaque modification and removal in peripheral artery disease, utilizing laser energy for tissue ablation.	[101]
6	Sterling PTA Balloon Catheter (Boston Scientific, Marlborough, MA, USA)	The Sterling PTA balloon catheter features a porous balloon design for percutaneous transluminal angioplasty (PTA) procedures, allowing for controlled dilation of the vessel lumen.	[102]
7	Thunderhawk™ RX PTA Balloon Dilatation Catheter (Abbott Vascular, Santa Clara, CA, USA)	The Thunderhawk RX PTA balloon catheter incorporates a microporous balloon design to dilate stenotic lesions in peripheral arteries.	[103]
8	UltraScore™ Focused Force PTA Balloon Dilatation Catheter (Becton, Dickinson, and Company, Franklin Lakes, NJ, USA)	The UltraScore PTA balloon catheter utilizes a microporous balloon design with focused force technology for optimal plaque dilation and lesion treatment.	[104]
9	Conquest™ High-Pressure PTA Balloon Dilatation Catheter (Medtronic, Dublin, Ireland),	The Conquest High-Pressure PTA Balloon Catheter features a porous balloon design for precise dilatation of stenotic lesions in peripheral arteries, especially in challenging cases.	[105]
10	Powerflex Pro™ PTA Balloon Dilatation Catheter (Cordis, a Cardinal Health company, Milford, OH, USA)	The Powerflex Pro PTA Balloon Catheter incorporates a porous balloon design with a low-profile shaft for enhanced deliverability and precise lesion treatment in peripheral arteries.	[106]

**Table 6 life-14-00451-t006:** Mechanisms involved in endovascular local drug release.

#	Mechanism	Description	Ref.
1	Diffusion	Drug molecules are released from the delivery system into the surrounding tissue through passive diffusion, driven by concentration gradients.	[115]
2	Degradation	The delivery system undergoes degradation over time, releasing encapsulated or bound drugs.	[116]
3	Chemical reactions	Specific chemical reactions, such as hydrolysis or enzymatic cleavage of bonds in the delivery system, can trigger drug release.	[117]
4	Swelling	Hydrogels or other polymer-based systems can swell in response to environmental stimuli (e.g., pH and temperature), leading to drug release.	[118]
5	External triggers	External stimuli, such as light, temperature, or magnetic fields, can trigger drug release from responsive materials.	[119]
6	Electrostatic interactions	Drug release can be modulated by electrostatic interactions between charged drug molecules and the delivery system, allowing for controlled release.	[120]
7	Osmotic pressure	Osmotic pressure gradients across semipermeable membranes can drive drug release from osmotically controlled delivery systems.	[121]
8	Mechanical activation	Mechanical forces applied to the delivery system (e.g., compression and stretching) can trigger drug release.	[122]
9	pH-responsive systems	pH-sensitive polymers or hydrogels can undergo conformational changes in response to local pH variations, leading to controlled drug release.	[123]
10	Magnetic targeting	Magnetic nanoparticles incorporated into the delivery system can be guided to the target site using an external magnetic field, enabling spatially controlled drug release.	[124]

**Table 7 life-14-00451-t007:** Future directions of endovascular drug delivery.

#	Direction	Description	Ref.
1	Nanotechnology-based delivery systems	Advancements in nanotechnology are expected to develop more precise and targeted drug delivery systems, including nanoparticles, liposomes, and dendrimers.	[213]
2	Bioresorbable stents with drug elution	Bioresorbable stents with incorporated drug-eluting capabilities are being explored to overcome the limitations of permanent metallic stents, potentially reducing the risk of late thrombosis and facilitating vessel healing.	[214]
3	Gene therapy	Gene therapy-based approaches for endovascular drug delivery aim to introduce therapeutic genes directly into target tissues to treat vascular diseases, such as restenosis and atherosclerosis.	[215]
4	Biomimetic materials	Biomimetic materials, inspired by natural extracellular matrix components, are being developed to enhance biocompatibility and promote tissue regeneration while delivering therapeutic agents.	[216]
5	Smart drug delivery systems	Integrating innovative materials and stimuli-responsive platforms into endovascular drug delivery systems allows for on-demand drug release triggered by specific environmental cues, such as pH, temperature, or biomarkers.	[217]
6	Personalized medicine approaches	Utilizing patient-specific data, such as genetic information and imaging diagnostics, to tailor endovascular drug delivery strategies for improved efficacy and patient outcomes.	[218]
7	Microfluidic-based delivery systems	Utilizing microfluidic technology to design miniaturized drug delivery devices capable of delivering precise doses of therapeutic agents to target sites within the vasculature.	[219]
8	Immuno-engineering	Engineering drug delivery systems that can modulate the immune response in vascular diseases, such as inflammation, atherosclerosis, and restenosis.	[220]
9	Remote-controlled drug delivery systems	Developing remote-controlled drug delivery systems that can be activated externally (e.g., using magnetic fields, ultrasound, or light) to trigger drug release at the desired location.	[221]
10	Bioactive coatings for implantable devices	Incorporating bioactive coatings on implantable devices to promote endothelialization, reduce thrombogenicity, and enhance the long-term performance of endovascular drug delivery systems.	[222]
11	Exosome-mediated drug delivery	Utilizing exosomes, natural nanovesicles secreted by cells, as carriers for targeted drug delivery to treat vascular diseases while minimizing off-target effects.	[223]

**Table 8 life-14-00451-t008:** Perivascular devices for drug delivery.

#	Device	Description	Ref.
1	Drug-eluting collagen wraps	Drug-eluting collagen wraps are biocompatible materials impregnated with therapeutic agents and wrapped around blood vessels during surgical procedures, enabling controlled local drug release to prevent restenosis or promote healing.	[229]
2	Perivascular drug-eluting mesh devices	Perivascular drug-eluting mesh devices are biodegradable or non-biodegradable meshes coated or impregnated with therapeutic agents and wrapped around blood vessels to provide sustained drug delivery locally, reducing neointimal hyperplasia or inhibiting thrombosis.	[230]
3	Perivascular drug delivery catheters	Perivascular drug delivery catheters are catheter-based devices equipped with drug-eluting balloons or porous membranes that can be placed around blood vessels to deliver therapeutic agents locally, targeting specific sites of vascular disease.	[231]
4	Perivascular drug-eluting hydrogels	Perivascular drug-eluting hydrogels are injectable or implantable materials placed around blood vessels to deliver therapeutic agents locally, providing sustained release and reducing restenosis or promoting vascular healing.	[232]
5	Perivascular drug-eluting stents	Perivascular drug-eluting stents are devices implanted around blood vessels to provide sustained drug delivery locally, preventing neointimal hyperplasia or inhibiting thrombosis while maintaining blood flow.	[233]
6	Perivascular drug-eluting patches	Perivascular drug-eluting patches are adhesive patches or wraps applied externally to blood vessels to deliver therapeutic agents locally, providing sustained drug release and targeting specific sites of vascular disease.	[234]
7	Perivascular drug-eluting coatings	Perivascular drug-eluting coatings are applied onto blood vessel surfaces or stents to provide localized drug delivery, reducing neointimal hyperplasia, inhibiting thrombosis, or promoting vascular healing.	[25]
8	Perivascular drug-eluting scaffolds	Perivascular drug-eluting scaffolds are three-dimensional structures placed around blood vessels to provide mechanical support and localized drug delivery, preventing restenosis or promoting vascular regeneration.	[235]
9	Perivascular drug-eluting gels	Perivascular drug-eluting gels are injectable or implantable materials placed around blood vessels to deliver therapeutic agents locally, providing sustained release and reducing restenosis or promoting vascular healing.	[236]
10	Perivascular drug-eluting films	Perivascular drug-eluting films are thin polymer films applied around blood vessels to provide controlled drug delivery, targeting specific sites of vascular disease and minimizing systemic side effects.	[237]

**Table 9 life-14-00451-t009:** Targeted endovascular drug delivery devices.

#	Method	Description	Ref.
1	Drug-eluting stents (DES)	Drug-eluting stents are implanted within blood vessels to prevent restenosis after angioplasty. They release drugs locally to inhibit neointimal hyperplasia and promote vascular healing.	[243]
2	Drug-coated balloons (DCBs)	Drug-coated balloons deliver antiproliferative drugs to the vessel wall during balloon angioplasty, aiming to prevent restenosis by inhibiting smooth muscle cell proliferation.	[244]
3	Drug-eluting beads (DEBs)	Drug-eluting beads are microspheres loaded with chemotherapeutic agents for targeted delivery to hepatic tumors via the hepatic artery, minimizing systemic toxicity.	[245]
4	Thrombolytic drug delivery systems	Thrombolytic agents are administered intravenously or intra-arterially to dissolve blood clots within the vasculature, restoring blood flow and preventing ischemic complications.	[246]
5	Intravenous drug delivery via nanoparticles	Through intravenous administration, nanoparticles can be engineered to deliver drugs to specific sites within the vascular system, such as atherosclerotic plaques or tumor vasculature.	[247]
6	Endovascular embolization	Endovascular embolization involves the delivery of embolic agents, such as microspheres or coils, into blood vessels to occlude blood flow, treat vascular malformations, or target tumors.	[248]
7	Intravascular drug-eluting microspheres	Drug-eluting microspheres are tiny beads loaded with therapeutic agents, such as chemotherapy drugs or radioisotopes, and delivered directly into the bloodstream via catheterization for targeted treatment of liver cancer or other vascular tumors.	[249]
8	Intravascular drug-eluting hydrogels	Drug-eluting hydrogels are injectable or implantable materials that can be delivered directly into blood vessels and solidify in situ, releasing therapeutic agents locally to prevent restenosis or treat vascular diseases.	[250]
9	Intravascular drug-eluting microbubbles	Drug-eluting microbubbles are gas-filled lipid or polymer microspheres that can be injected intravenously and targeted to specific vascular sites using ultrasound, enhancing drug delivery and therapeutic efficacy.	[251]
10	Intravascular gene delivery vectors	Viral or non-viral vectors can be used for endovascular gene therapy by delivering therapeutic genes directly into blood vessels to modulate vascular function, promote angiogenesis, or treat genetic vascular disorders.	[252]
11	Intravascular nanofibrous scaffolds	Nanofibrous scaffolds can be delivered endovascularly and serve as platforms for local drug delivery, tissue engineering, or promoting vascular regeneration by providing structural support and controlled release of bioactive molecules.	[253]
12	Intravascular magnetic drug targeting	Magnetic nanoparticles loaded with therapeutic agents can be delivered intravenously and targeted to specific vascular sites using external magnetic fields, enabling site-specific drug delivery and enhanced therapeutic efficacy.	[254]

## Data Availability

Data sharing is not applicable.

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
