# Peer review of "Endovascular Drug Delivery"

_life, 2024, doi:10.3390/life14040451_

Round 1

Reviewer 1 Report

Comments and Suggestions for Authors

1. It would be beneficial for the manuscript if the introduction provided a clear overview and focus of the review. This would help readers grasp the main themes and objectives from the outset. Additionally, enhancing the connections and logic between paragraphs would improve the coherence of the paper.

2. Consider refining the abstract to ensure conciseness while also tailoring it to the targeted audience. A succinct abstract can effectively convey the essence of the research and attract the interest of readers.

3. In line 91-93, it's advisable not to include references for statements that the author intends to discuss further in the review. 

4. Please remember to include the reference in line 103 to maintain the integrity of the citation process.

5. Adding a brief description of the mechanism of DES in the first paragraph of section 2.1 could enhance the flow of the content and provide clarity to readers. Same to the other sections. 

6. When discussing tables in the review, consider including a brief explanation of how readers can benefit from the information presented in the tables. This will help readers understand the relevance and utility of the tables in their research.

7. Ensure smooth transitions between sections to maintain the coherence and readability of the manuscript. Transition statements can guide readers through the logical progression of the paper.

8. Clarify the distinction between section 2.6 and section 2.9, and consider whether the 18 subsections should belong to section 2 based on their thematic similarity.

9. Section 3, focusing on future directions, is a pivotal part of the manuscript. Enhancing the logical flow and coherence of this section will improve its effectiveness in communicating key ideas to readers.

10. Reflect on the key messages you want to convey in your manuscript and consider adopting an engaging storytelling approach. By crafting a compelling narrative, you can effectively communicate the significance of your research and its potential impact on the field.

Comments on the Quality of English Language

N/A

Author Response

Comment 1. It would be beneficial for the manuscript if the introduction provided a clear overview and focus of the review. This would help readers grasp the main themes and objectives from the outset. Additionally, enhancing the connections and logic between paragraphs would improve the coherence of the paper.

 Response 1: As suggested the following text was added: Overall, this review is focused on endovascular drug delivery methods and devices used in atherosclerotic disease therapy. Firstly, intraluminal drug delivery devices are discussed. A short overview of atherosclerotic diseases and the hyperplastic vascular response is discussed. The impact of vascular and atherosclerotic structures on drug retention is also presented. Bioresorbable stems are given special attention. Porous and Mi-croporous balloons are presented, and some commercially available devices are named. The mechanisms controlling local drug release are synthesized. The device-based endo-vascular drug delivery strategies are detailed. Also, devices for vascular drug delivery and the release mechanism are discussed. First-generation and second-generation DES(drug-eluting stents ) are overviews. Bioresorbable scaffolds, drug-coated balloons, future directions: transition toward lesion-specific drug delivery, deployable coatings, polymer-free coated stents, and polymer-free coated stents are presented. The discussion section includes an overview of the clinical implications of lesion-specific intervention. Furthermore, lesion-specific intervention and future drug delivery, looking toward the horizon, are presented. Lastly, extraluminal drug delivery devices, perivascular bio-materials, systemic drug administration, and targeted intravenous drug delivery are described.

Comment 2. Consider refining the abstract to ensure conciseness while also tailoring it to the targeted audience. A succinct abstract can effectively convey the essence of the research and attract the interest of readers.

 Response 2: The abstract was rewritten, and the following text was added: Drug-eluting stents (DES) and balloons revolutionize atherosclerosis treatment by targeting hyperplastic tissue responses through effective local drug delivery strategies. This review examines approved and emerging endovascular devices, discussing drug release mechanisms and their impacts on arterial drug distribution. It emphasizes the crucial role of drug delivery in modern cardiovascular care and highlights how device technologies influence vascular behavior based on lesion morphology. The future holds promise for le-sion-specific treatments, particularly in the superficial femoral artery, with recent CE-marked devices showing encouraging results. Exciting strategies and new patents focus on local drug delivery to prevent restenosis, shaping the future of interventional outcomes. In summary, as we navigate the ever-evolving landscape of cardiovascular intervention, it becomes increasingly evident that the future lies in tailoring treatments to the specific characteristics of each lesion. By leveraging cutting-edge technologies and harnessing the potential of localized drug delivery, we stand poised to usher in a new era of precision medicine in vascular intervention.

Comment 3. In line 91-93, it's advisable not to include references for statements that the author intends to discuss further in the review. 

 Response 3: corrected

Comment 4. Please remember to include the reference in line 103 to maintain the integrity of the citation process.

 Response 4: corected

Comment 5. Adding a brief description of the mechanism of DES in the first paragraph of section 2.1 could enhance the flow of the content and provide clarity to readers. Same to the other sections. 

 Response 5: the following text was added: Drug-eluting stents (DES) are small mesh-like tubes inserted into narrowed or blocked blood vessels to restore blood flow. They work by combining the mechanical support of a traditional stent with a drug delivery system. The stent scaffold helps keep the artery open while the drug, typically an anti-proliferative or anti-inflammatory agent, is released gradually from the stent coating into the surrounding tissue. This drug helps prevent the re-narrowing of the artery, known as restenosis, by inhibiting excessive tissue growth or inflammation that can occur in response to the stent placement. By combining mechanical support with targeted drug delivery, DES effectively reduces the risk of reste-nosis and improves long-term outcomes for patients undergoing coronary or peripheral vascular interventions.

For section 2.4 the following text was added: Bioresorbable stents are designed to dissolve and be absorbed by the body gradually over time. They are typically made from materials such as polylactic acid or magnesium alloy. When implanted in a narrowed or blocked blood vessel, bioresorbable stents pro-vide temporary support to keep the artery open, similar to traditional stents. Over time, the stent material gradually breaks into biocompatible byproducts, allowing the artery to return to its natural state without a permanent implant. This gradual dissolution reduces the risk of long-term complications associated with permanent stents, such as late stent thrombosis or vessel re-narrowing.

Additionally, as the stent dissolves, it allows the artery to regain its ability to constrict and dilate in response to changes in blood flow, which may promote better long-term vascular health

For 2.5 the followig text was added: Porous and microporous balloons are used to open narrowed or blocked blood ves-sels in angioplasty procedures. When inflated, they exert radial force against the vessel walls, which helps compress the plaque and widen the artery. The pores in these balloons allow for the delivery of drugs or contrast agents directly to the vessel wall during infla-tion. This localized drug delivery can help prevent restenosis or other complications by targeting specific areas of the vessel where the balloon is in contact. The microporous structure may also enhance drug absorption and distribution within the vessel wall, po-tentially improving treatment outcomes. Overall, porous and microporous balloons com-bine mechanical dilation with targeted drug delivery to optimize the effectiveness of an-gioplasty procedures

For 2.7.1 the following text was added: Durable adherent coatings are used in medical devices, such as stents, to improve their performance and longevity. These coatings adhere tightly to the surface of the device and are designed to withstand the harsh environment within the body. They typically consist of biocompatible materials that are resistant to degradation and provide a smooth surface to reduce the risk of clot formation or tissue irritation. Durable adherent coatings help enhance the biocompatibility of medical devices, promote better tissue integration, and minimize adverse reactions within the body, ultimately improving patient outcomes

For 2.7.2 the following text was added: Biodegradable adherent coatings are designed to degrade within the body over time gradually. These coatings are applied to medical devices, such as stents, to provide tem-porary protection and promote healing while the device is in place. As the coating breaks down, it releases any incorporated drugs or therapeutic agents, which are gradually ab-sorbed by the surrounding tissue. This controlled release helps prevent complications and promotes the desired therapeutic effect. Once the coating has fully degraded, it is naturally eliminated from the body, leaving behind a fully functional medical device. Biodegradable adherent coatings offer the benefits of targeted drug delivery and reduce long-term risks associated with permanent coatings, making them valuable in various medical applica-tions.

For 2.10 : Various coating and excipient formulations are hypothesized to influence the result-ing tissue response, and comprehensive reviews on durable, biodegradable, or deployable coating technologies are readily accessible

For the 2.11 the following text was added: First-generation drug-eluting stents (DES) consist of a metal stent scaffold coated with a polymer containing an anti-proliferative drug. When deployed in a narrowed or blocked blood vessel, the stent provides mechanical support while the drug-coated polymer grad-ually releases the medication into the surrounding tissue. This drug inhibits the growth of smooth muscle cells, reducing the risk of restenosis or re-narrowing of the artery. Over time, the polymer coating may degrade or be absorbed by the body, leaving behind the bare metal stent. Despite their efficacy in reducing restenosis rates compared to bare metal stents, first-generation DES are associated with potential long-term complications such as delayed healing and inflammation due to the presence of durable polymer coatings

For 2.12 the following text was added: Second-generation drug-eluting stents (DES) improve upon first-generation designs by addressing limitations such as delayed healing and inflammation associated with durable polymer coatings. These stents typically feature thinner struts and more biocom-patible or bioabsorbable polymer coatings. The drug release mechanism remains similar, with an anti-proliferative drug gradually released from the polymer coating to inhibit smooth muscle cell proliferation and reduce the risk of restenosis. Additionally, sec-ond-generation DES may incorporate modifications to enhance deliverability, flexibility, and conformability within the vessel, improving procedural outcomes and patient com-fort. Overall, second-generation DES aim to provide effective and safer treatment options for patients undergoing coronary or peripheral vascular interventions.

For 2.13 the following text was added: Bioresorbable scaffolds are temporary implants used in coronary artery interventions. Made from biodegradable materials like polylactic acid, they provide structural support to keep a narrowed artery open, similar to traditional metal stents. Over time, the scaffold gradually dissolves and is absorbed by the body, allowing the artery to return to its natu-ral state. This gradual absorption reduces the risk of long-term complications associated with permanent stents, such as late stent thrombosis. Additionally, as the scaffold dis-solves, the artery can regain its ability to constrict and dilate, potentially improving long-term vascular health

For 2.14 the following text was added: Drug-coated balloons (DCBs) are medical devices used in angioplasty procedures to treat narrowed or blocked blood vessels. These balloons are coated with a drug, typically an anti-proliferative or anti-inflammatory agent. When the balloon is inflated within the narrowed artery, the drug is transferred directly to the vessel wall, inhibiting smooth muscle cell proliferation and reducing inflammation. This localized drug delivery helps prevent restenosis or re-narrowing of the artery following the procedure. DCBs offer a tar-geted and practical approach to treating vascular disease, particularly in cases where tra-ditional balloon angioplasty alone may not be sufficient.

For 2.16 the following text was added: Drug-deployable coatings are medical coating applied to devices used in vascular interventions, such as stents or balloons. These coatings contain drugs or therapeutic agents released upon device deployment within the blood vessel. When the device is ex-panded or inflated, the coating comes into contact with the vessel wall, triggering the re-lease of the incorporated drug. This localized drug delivery helps prevent restenosis or other complications by targeting specific areas of the vessel where the device is in contact. Drug-deployable coatings offer a controlled and targeted approach to delivering medica-tions directly to the treatment site, improving the efficacy of vascular interventions

For 2.17 the following text was added:

Polymer-free coated stents are a type of stent used in coronary artery interventions that do not rely on a polymer coating to deliver drugs. Instead, these stents use a proprietary surface treatment to directly attach drugs or therapeutic agents to the stent surface. When deployed in a narrowed or blocked blood vessel, the stent releases the drugs into the surrounding tissue, helping to inhibit smooth muscle cell proliferation and reduce inflammation. This localized drug delivery helps prevent restenosis or re-narrowing of the artery following the procedure. Polymer-free coated stents offer an alternative approach to drug delivery that may reduce the risk of long-term complications associated with traditional polymer-coated stents.

For 2.18 the following statement was added: Drug-filled stents are used in vascular interventions that contain a reservoir or coat-ing filled with medication. These stents release the drug directly into the surrounding tis-sue upon deployment, helping to inhibit smooth muscle cell proliferation and reduce in-flammation. This localized drug delivery helps prevent restenosis or re-narrowing of the artery following the procedure. Drug-filled stents offer a practical approach to delivering medications directly to the treatment site, improving the efficacy of vascular interventions while reducing the risk of systemic side effects associated with oral medica-tions

For 3.3 the folowing text was added: Extraluminal drug delivery devices and perivascular biomaterials are designed to deliver medication around blood vessels rather than directly into them. These devices typically involve the placement of drug-eluting implants or coatings around the outside of the blood vessel, targeting the tissue surrounding the vessel rather than the vessel lumen itself. The medication is gradually released from these devices, diffusing into the sur-rounding tissue to inhibit smooth muscle cell proliferation, reduce inflammation, or achieve other therapeutic effects. This approach can be beneficial in cases where direct in-traluminal drug delivery is not feasible or practical, providing a targeted and localized treatment option for vascular diseases.

Comment 6. When discussing tables in the review, consider including a brief explanation of how readers can benefit from the information presented in the tables. This will help readers understand the relevance and utility of the tables in their research.

 Response 6: Added description to Table 1, Table 2, and Table 9. In Tables 3,4 and 5,6,7,8, the description was already included in the text.

Comment 7. Ensure smooth transitions between sections to maintain the coherence and readability of the manuscript. Transition statements can guide readers through the logical progression of the paper.

 Response 7: Corrected

Comment 8. Clarify the distinction between section 2.6 and section 2.9, and consider whether the 18 subsections should belong to section 2 based on their thematic similarity.

 Response 8: paragraph 2.6 and 2.9 are disctinct futures. Paragraph 2.89 was renamed. Also it was expended. The folowing text was added: The mechanisms for controlling drug release in drug-eluting stents (DES) can be broadly classified into physical and chemical categories. Physical control involves various methods such as drug diffusion through a durable polymer coating, release from a dis-solving or eroding coating matrix, drug-specific binding via ion exchange or immobilized tissue antibodies, or utilizing osmotic pressure differences to propel the drug into the vascular wall. These mechanisms play a predominant role in DES, with manufacturing parameters like polymer formulation, coating thickness, and exposed stent surfaces di-rectly influencing the predictability of release rates. On the other hand, chemical release control involves breaking the bonds between the drug and its carrier coating to release the drug. However, this method requires chemical modification of the drug to bind to its car-rier, resulting in drug release in a prodrug form, which has not gained significant traction in the field of DES. Several major mechanisms are utilized to control drug release in DES, including polymer matrix, diffusion, erosion, ion exchange, and biodegradable coatings. In many DES, a polymer matrix serves as a carrier for the drug, where the drug is dis-solved or dispersed within the polymer, gradually releasing in a controlled manner as the polymer degrades over time. Common polymers used include poly(lactic-co-glycolic acid) (PLGA), polyethylene-co-vinyl acetate (PEVA), and poly(styrene-b-isobutylene-b-styrene) (SIBS). In diffusion-controlled drug release, molecules move through the polymer matrix via diffusion gradients, with factors like molecular weight, size, and concentration gradi-ent influencing the release rate. The properties of the polymer matrix, such as porosity and tortuosity, can be adjusted to modulate the release rate. Erodible polymer matrices are em-ployed in some DES, which degrade over time in response to physiological conditions (e.g., pH, enzymes) or mechanical stress, releasing the encapsulated drug as the polymer degrades. The erosion rate can be tailored by adjusting the polymer's chemical composi-tion and molecular weight. Certain DES incorporate ion-exchange mechanisms for drug release, where the drug is bound to charged groups within the polymer matrix. Upon ex-posure to bodily fluids, ions in the fluid compete with the drug for binding sites on the polymer, leading to drug release. Modifying the ionic properties of the polymer can control the release rate. Additionally, DES may feature biodegradable coatings that encapsulate the stent and drug, gradually degrading over time and releasing the drug into the sur-rounding tissue, with the composition and thickness of the coating influencing release ki-netics. Advanced DES may also include mechanisms responsive to external stimuli like temperature, light, or magnetic fields to trigger drug release. These stimuli-responsive materials undergo conformational changes or degradation in response to specific stimuli, enabling controlled drug release

Comment 9. Section 3, focusing on future directions, is a pivotal part of the manuscript. Enhancing the logical flow and coherence of this section will improve its effectiveness in communicating key ideas to readers..

Response 9:Most of the parts in Section 3 habe ben rephrases and shortend in order to  more concice Some part of section 3 were entirely rewitten: 31 32 3.5 .

The folowing text was added: The anticipated clinical transition toward lesion-specific intervention necessitates a holis-tic approach that is not reliant on a single device or intervention adjustment. It entails simultaneous modifications in lesion characterization methods, device selection, and drug formulations. Moreover, purposefully designed clinical trials are crucial to validate the significance of lesion-specific characterization in clinical practice [196,197].

Lesion-specific characterization through medical imaging is essential for advancing tailored interventions. Intravascular imaging, particularly virtual histology by intravas-cular ultrasound (VH-IVUS), allows detailed insights into plaque morphology. VH-IVUS classifies tissue into fibrotic, fibro-fatty, calcific, and necrotic core categories, validated for coronary lesions. It also correlates lesion phenotype with hyperplastic development in coronary drug-eluting stents (DESs). Near-infrared spectroscopy enhances lipid detection alongside VH-IVUS. Optical coherence tomography provides high-resolution imaging, aiding vascular drug delivery and identifying calcific developments affecting delivery. Recently, optical coherence tomography has incorporated virtual histology-like features. It is desirable to develop similar micromorphology characterization from noninvasive an-giography, given its widespread use in clinical practice, albeit with existing qualitative scores on calcium burden. [198,199,200].

Imaging is vital in quantifying lesion morphology and is expected to be pivotal in an era focused on lesion-specific interventions. Numerous examples illustrate the effective integration of imaging into clinical practice for identifying vulnerable plaques, assessing acute and long-term treatment efficacy, and evaluating regional lesion status before drug-eluting stent (DES) implantation. A recent meta-analysis underscored improved pa-tient survival by incorporating intravascular imaging and lesion assessment before DES implantation. In the context of a lesion-specific transition, quantifying lesion morphology through medical imaging could inform the selection of devices and drugs for interven-tional treatment. For example, lipid-rich plaques might suggest sirolimus-based delivery due to enhanced tissue-specific paclitaxel binding displacement.

Conversely, hemorrhagic or thin fibrotic structures might indicate structural instabil-ity, necessitating rigid stent-based delivery. [201,202]. To definitively assess the benefits of morphology-resolving imaging, image-derived lesion morphology should be incorporated as a focal point in clinical evaluations of drug delivery devices. This entails monitoring volumetric content, anatomic positioning (superficial vs. deep), and morphologic region-ality (local vs. circumferential) concerning clinical outcomes. [203,204].

Similarly, image-based lesion characterization could assist in selecting appropriate lesion preparation strategies, potentially enhancing subsequent retention of delivered drugs. Recent review literature emphasizes that identifying highly calcific structures through imaging could justify pre-procedural removal or modification to improve drug permeation. Likewise, detecting lipid-rich areas could offer insights into mechanical sta-bility, displacement of critical tissue-binding sites, or even reveal the actual lesion length in complex phenotypes. Overall, lesion preparation could help standardize vascular re-sponse and drug retention by adapting to changing morphologic structures, with imaging playing a pivotal role in elucidating definitive vascular mechanisms driving such behav-ior. While comprehensive clinical data on the impact of lesion preparation on adjunctive drug-eluting technologies is still pending, ongoing trials like Disrupt CAD/PAD I–III and ECLIPSE hold promise for generating such data soon. [205,206,207,208].

Lesion-specific intervention extends to the choice of drug and device. Imaging pro-vides insight into morphological characteristics, influencing the selection of intervention strategies. For drug choice, there is ongoing debate in the interventional field. Levin et al. reviewed that transluminal retention relies on drug-specific binding sites, with siroli-mus-based drugs targeting mammalian targets of rapamycin receptors distributed more evenly through the vessel wall. In contrast, paclitaxel primarily targets tubulin in the sub-intimal and adventitial layers. Device-induced intimal disruption or atherosclerotic dis-placement of binding sites may be more detrimental to paclitaxel-based delivery.

Preclinical studies have demonstrated that sirolimus binding is less sensitive to le-sion complexity, whereas paclitaxel-based partition coefficients decline rapidly with in-creasing lipid content. [209,210,211]. Complex lesions with an active core and distinct li-pidic components may benefit from sirolimus-based delivery. Long-term retention can be enhanced by prioritizing sirolimus formulations with high partition coefficients, such as the more hydrophobic zotarolimus or everolimus compared to sirolimus. The preference for paclitaxel delivery remains in cases of high medial-to-deep calcification, where pro-found permeation barriers shift binding to subintimal spaces. Assuming minimal subin-timal disruption from the implanted or inflated device, the higher partition coefficient of paclitaxel could lead to a more efficient antiproliferative effect. In contemporary practice, paclitaxel delivery also appears favored in calcified peripheral lesions. [212,213,214]. However, lesion preparation could present a more feasible alternative to address the spe-cific environment before intervention in such scenarios. Experimental and clinical valida-tion of paclitaxel versus sirolimus in such contexts is still pending. A shift towards le-sion-specific approaches requires conclusive clinical evidence guided by purposefully designed clinical trials. [215,216].

Besides the influence of lesion complexity on drug formulation, the visible lesion morphology also guides the selection of interventional devices. The difference in delivery mode between drug-coated balloons (DCBs) and drug-eluting stents (DESs) or bioresorba-ble vascular scaffolds (BVSs) is crucial to consider to lesion complexity. DCBs rely on short-term mechanical deposition during inflation, facilitating delivery into soft, hema-toma-rich, fatty lesions (e.g., in-stent restenosis tissue) but obstructing delivery into fi-bro-calcific or calcific lesions. Coating micromorphology may help overcome more rigid intimal entities, with microneedle configurations inducing higher contact pressure and coating transfer than amorphous coating equivalents. In highly calcific lesions, stent-based intervention and appropriate vessel preparation are preferred. Atherectomy reduces superficial calcium through physical grinding, while contemporary lithotrip-sy-based techniques address deep calcium. Vessel preparation may rectify divergent re-sults stemming from heterogeneous lesion phenotypes, yet conclusive large-scale clinical outcome results in this area are still pending. [217,218,219].

Considering the heterogeneity of atherosclerotic lesions, we foresee lesion-specific in-tervention as transitioning towards tailored vessel preparation techniques and delivery devices guided by lesion characteristics rather than adhering to a one-size-fits-all ap-proach [220,221].

 The shift towards lesion-specific drug delivery intervention benefits from continu-ous updates and refinements driven by advanced computational and scientific evalua-tions. High-fidelity imaging enables patient-specific modeling for predictive assessments of intervention outcomes. Computational simulations evaluate factors governing hemo-dynamic, structural, and drug responses. Integration of pharmacokinetic and pharmaco-dynamic profiles enhances understanding of drug delivery mechanisms. Recent ad-vancements aim to include lesion-specific components in simulations, highlighting the ongoing role of computational modeling in establishing guidelines. [222,223,224].

Nonhuman animal modeling is crucial for establishing frameworks for le-sion-specific intervention. Animal models offer a controlled study environment, allowing isolated evaluations while retaining the complexity of the cardiovascular system. They have been central to developing local drug delivery devices, considering lesion-specific aspects. Understanding that animal and computational modeling should not precede clinical evaluations is essential. Instead, developing local drug delivery devices should involve iterative evaluations encompassing theoretical, computational, animal, and clini-cal assessments. [225,226]. The following section presents the future of lesion-specific in-terventions and endovascular drug delivery perspectives.

 Extraluminal drug delivery devices and perivascular biomaterials are designed to deliver medication around blood vessels rather than directly into them. These devices typically involve the placement of drug-eluting implants or coatings around the outside of the blood vessel, targeting the tissue surrounding the vessel rather than the vessel lumen itself. The medication is gradually released from these devices, diffusing into the sur-rounding tissue to inhibit smooth muscle cell proliferation, reduce inflammation, or achieve other therapeutic effects. This approach can be beneficial in cases where direct in-traluminal drug delivery is not feasible or practical, providing a targeted and localized treatment option for vascular diseases. Periadventitial drug delivery via a perivascular wrap offers an alternative approach. In a study by Kelly et al., ethylene vinyl acetate peri-vascular wraps loaded with paclitaxel were evaluated in a porcine model of arteriovenous graft stenosis. Anastomoses treated with paclitaxel-loaded wraps during surgery showed reduced luminal stenosis compared to untreated graft-vein anastomoses (0.17% in the paclitaxel group vs. 37.90% stenosis in the control group[242,243]. Poly(ε-caprolactone) (PCL), a biocompatible and biodegradable polymer, has been utilized for delivering paclitaxel and rapamycin to the vessel wall. Extensive in vitro and in vivo studies have led to FDA approval for various medical drug delivery devices com-posed of PCL. In a mouse femoral artery injury model, PCL cuffs loaded with paclitaxel or rapamycin and control cuffs were placed around injured femoral arteries. After three weeks, paclitaxel and rapamycin-eluting PCL cuffs reduced intimal thickening by 76% and 75%, respectively, with localized delivery and no observed adverse systemic effects. [244].

Although showing promising results in animal models, the transition of this tech-nology to human clinical trials has been sluggish. Only one company, VesselTek Biomed-ical (Chicago, IL), is currently nearing human trials. They are developing a drug-eluting perivascular wrap (VTek-RA wrap) made from poly(diol citrate). [245].

Utilizing tissue-specific targeted systemic treatments for local drug delivery shows promise. In this approach, agents are administered systemically but include tis-sue-specific tags that guide them to the injured vessel following vascular intervention. De-glau et al. investigated a site-specific delivery system employing microspheres carrying therapeutic drugs. These microspheres consisted of reactive polyethylene glycol tagged with avidin. Using the Remedy microporous balloon, a balloon coated the injured artery with biotin. Avidin-coated microspheres were then intravenously administered, attaching to the biotin on the arterial wall and locally releasing the drug. This innovative approach has the potential to precisely deliver systemically injected anti-restenotic agents to the tar-get site, offering a more targeted and effective treatment strategy. [261].

Targeted drug delivery through systemic injection using microspheres represents an innovative method. These microspheres can be engineered to target proteins specifically expressed or upregulated after vascular injury. For example, researchers have developed microspheres or particles that directly target surface markers exposed following vascular injury, such as E- and P-selectin, ICAM-1, and VCAM-1. Similarly, nanoparticles can be modified to target and bind to specific proteins within the injured arterial wall. In a study by Chan et al., they designed a nanoparticle with a lipid core-shell interface between pol-ylactide-co-glycolic acid and polyethylene glycol polymers. This nanoparticle was loaded with paclitaxel and had peptides directed against collagen IV, aiming to bind to collagen IV in the basal lamina of the vessel wall exposed after endothelial denudation from me-chanical injury. Safety studies in rats showed no signs of toxicity. However, further clini-cal trials are necessary to assess the safety and efficacy of these innovative techniques in human subjects. [262]. Some examples are listed in the table below(Table 9), where the most used endovascular targeted devices are listed together with their implication in in-timal hyperplasia:

 Also, Figure1 and  2 were introduced.

Comment 10. Reflect on the key messages you want to convey in your manuscript and consider adopting an engaging storytelling approach. By crafting a compelling narrative, you can effectively communicate the significance of your research and its potential impact on the field.

Response 10: The paragraphs have been rewritten and rephrased to the whole manuscript to be more concise as suggested.

Reviewer 2 Report

Comments and Suggestions for Authors

It is an informative review and the readers can clearly apprehend all the efforts they have spent on this paper  However, it is too long and any differences from already published papers on similar topics were not distinctively apparent.

·       Need to be compared with those papers, such as  

Nezami, F. R.; Athanasiou, L. S.; Edelman, E. R. Chapter 28 - Endovascular Drug-delivery and Drug-elution Systems. In Biomechanics of Coronary Atherosclerotic Plaque; Ohayon, J., Finet, G., Pettigrew, R. I., Eds.: Academic Press: Massachusetts, 2021; Vol. 4, pp 595–631.

Tzafriri AR, Edelman ER. Endovascular Drug Delivery and Drug Elution Systems: First Principles. Interv Cardiol Clin. 2016 Jul;5(3):307-320. doi: 10.1016/j.iccl.2016.02.007. Epub 2016 Jun 21. PMID: 28582029; PMCID: PMC6661070.

·       Need some Figures to elucidate the drug release mechanisms.

Author Response

Comment 1: It is an informative review and the readers can clearly apprehend all the efforts they have spent on this paper  However, it is too long and any differences from already published papers on similar topics were not distinctively apparent.

Response1: The general recommendation guideline is that the revie must be longer than 4000 words. The review was designed to be as comprehensive as possible and to cover as many topics as possible. The purpose of the review is to summarize what is up to date and what the basic prices of a particular subject are.

 Comment 2:   Need to be compared with those papers, such as  Nezami, F. R.; Athanasiou, L. S.; Edelman, E. R. Chapter 28 - Endovascular Drug-delivery and Drug-elution Systems. In Biomechanics of Coronary Atherosclerotic Plaque; Ohayon, J., Finet, G., Pettigrew, R. I., Eds.: Academic Press: Massachusetts, 2021; Vol. 4, pp 595–631. Tzafriri AR, Edelman ER. Endovascular Drug Delivery and Drug Elution Systems: First Principles. Interv Cardiol Clin. 2016 Jul;5(3):307-320. doi: 10.1016/j.iccl.2016.02.007. Epub 2016 Jun 21. PMID: 28582029; PMCID: PMC6661070.

Response 2: Both references were consulted as suggested. Reference 2 was already included in the reference list ( reference 140.                Tzafriri AR, Edelman ER. Endovascular drug delivery and drug elution systems: first principles. In-terv Cardiiol Clin 2016;5:307–20.  )

Comment 3: Need some Figures to elucidate the drug release mechanisms.

Response3:  As suggested, Figure 1  and Figure 2 were  added to the manuscript. Also, some parts of the manuscript were rewritten.

Reviewer 3 Report

Comments and Suggestions for Authors

The manuscript is interesting and well written, but needs a minor correction.

1.     Page 5 line 191: this sentence is unclear. What is mean in calcium removal? Here we are talking about calcinate already present in the vessel. If we mean prevention of vascular calification by calcium removal, it should appear in the sentence. 

2.     Reference 20 not complete (2016; 44(2): 276-286)

3.     Reference 21 not presented

4.     Reference 52 not complete (2015; 66: B224-B225)

5.     Reference 65 not complete (2015; 11(5): 1039-46)

6.     References 104 and 106 not presented

7.     Reference 218 not complete (2011; 29(6): e54-66)

8.     Reference 229 not complete (2012; 14(5): 635-41)

Author Response

The manuscript is interesting and well written but needs a minor correction.

Comment1:     Page 5 line 191: this sentence is unclear. What is mean in calcium removal? Here we are talking about calcinate already present in the vessel. If we mean prevention of vascular calification by calcium removal, it should appear in the sentence.

Response 1: The phrase was rewritten as suggested: Vascular calcification, prevalent in peripheral arteries, significantly impacts drug permeability by creating an impenetrable structural diffusion barrier. Studies show that preventing vascular calcification by calcium removal increases drug diffusivity and absorption rate, with clinical evidence linking decreased treatment efficacy to increasing calcium.

Comment2:     Reference 20 not complete (2016; 44(2): 276-286)

Response 2: The reference was completed as suggested: 20.         Artzi N, Tzafriri AR, Faucher KM et al. Sustained Efficacy and Arterial Drug Retention by a Fast Drug Eluting Cross-Linked Fatty Acid Coronary Stent Coating. Ann Biomed Eng. 2016; 44(2): 276-286.

Comment 3:     Reference 21 not presented

Response 3: The reference was added: Kinam Park. Dual drug-eluting stent. J Control Release. 2012 Apr 10;159(1):1

Comment4:     Reference 52 not complete (2015; 66: B224-B225)

Response 4: The reference was completed as suggested: Tzafriri AR, Markham PM, Goshgarian J et al. TCT-554 Titratable drug delivery from drug filled stents. Journal of the American College of Cardiology 2015;66: B224-B225

Comment5:     Reference 65 not complete (2015; 11(5): 1039-46)

Response 5: The reference was complete as suggested: Van der Valk FM, van Wijk DF, Lobatto ME, et al. Prednisolone-containing liposomes accumulate in human atherosclerotic macrophages upon intravenous administration. Nanomedicine (Lond). 2015;11(5):1039-46.

Comment6:     References 104 and 106 not presented

Response 6: References 104 and 106 were added : 104.  Japanese Association of Cardiovascular Intervention and Therapeutics: Clinical expert consensus document on drug-coated balloons for coronary artery disease was produced by the Japanese Association of Cardiovascular Intervention and Therapeutics. Cardiovascular Intervention and Therapeutics : 2023; 38(1):1-14.

  1. Spiliopoulos S, Karamitros A, Reppas L, Brountzos E. Novel balloon technologies to minimize dissection of peripheral angioplasty. Expert Review of Medical Devices.2019, 16(7):581-588

Comment7:    Reference 218 not complete (2011; 29(6): e54-66)

Response7: The reference was corrreted as suggested: 218.          Gertz ZM, Wilensky RL. Local Drug Delivery for Treatment of Coronary and Peripheral Artery Dis-ease. Cardiovasc Ther. 2011; 29(6): e54-66.

Comment 8: Reference 229 not complete (2012; 14(5): 635-41)

Response8: The reference was corrected as suggested: 229.          Wohrle J. Drug-Coated Balloons for Coronary and Peripheral Interventional Procedures. Curr Cardiol Rep. 2012; 14(5): 635-41

Round 2

Reviewer 1 Report

Comments and Suggestions for Authors

I appreciate the significant improvements made to the manuscript by the authors. However, the review still needs to better highlight its key message. Given the dense information it contains which could greatly benefit young researchers in the field. I would recommend publication subject to the following revisions:

1. Please correct the error on line 15 of the abstract. Similar errors are present throughout the manuscript. I advise a thorough examination of the entire document to identify and rectify these issues.

2. Expand all abbreviations in the abstract for clarity.

3. I strongly suggest that the authors reconsider the manuscript's title if the primary focus is on devices.

4. Relocate figure legends to the bottom of the figures for consistency.

5. Pay attention to the sequencing and formatting of Figures 1 and 2; Figure 1 should not be presented after Figure 2. Additionally, higher-resolution and informative figures should be provided.

Comments on the Quality of English Language

n/a